# Rectifying Conformity Scores for Better Conditional Coverage

Vincent Plassier [* 1]   Alexander Fishkov [* 2 3]   Victor Dheur [* 4]   Mohsen Guizani [2]   Souhaib Ben Taieb [2 4]
Maxim Panov [2]   Eric Moulines [2 5]

## Abstract

We present a new method for generating confidence sets within the split conformal prediction framework. Our method performs a trainable transformation of any given conformity score to improve conditional coverage while ensuring exact marginal coverage. The transformation is based on an estimate of the conditional quantile of conformity scores. The resulting method is particularly beneficial for constructing adaptive confidence sets in multi-output problems where standard conformal quantile regression approaches have limited applicability. We develop a theoretical bound that captures the influence of the accuracy of the quantile estimate on the approximate conditional validity, unlike classical bounds for conformal prediction methods that only offer marginal coverage. We experimentally show that our method is highly adaptive to the local data structure and outperforms existing methods in terms of conditional coverage, improving the reliability of statistical inference in various applications.

## 1. Introduction

The widespread deployment of AI models emphasizes the need for reliable uncertainty quantification (Gruber et al., 2023). Although highly flexible in capturing complex statistical dependencies, these models can produce unreliable or overly confident predictions (Nalisnick et al., 2018). Conformal prediction (CP; Vovk et al. (2005); Shafer & Vovk (2008)) offers a robust, distribution-free framework for predictions with finite-sample validity guarantees (Angelopoulos et al., 2023; 2024).

*Equal contribution [1]Lagrange Mathematics and Computing Research Center [2]Mohamed bin Zayed University of Artificial Intelligence [3]Skolkovo Institute of Science and Technology [4]University of Mons [5]École Polytechnique. Correspondence to: Maxim Panov <maxim.panov@mbzuai.ac.ae>.

*Proceedings of the $42^{nd}$ International Conference on Machine Learning*, Vancouver, Canada. PMLR 267, 2025. Copyright 2025 by the author(s).

Classical CP approaches guarantee marginal validity but fail to ensure the more desirable property of conditional validity, which customizes prediction regions to specific covariates. Prior studies have shown constructing meaningful prediction regions with exact conditional validity is infeasible without additional distributional assumptions (Vovk, 2012; Lei & Wasserman, 2014; Foygel Barber et al., 2021). Consequently, current research emphasizes developing conformal methods that maintain marginal validity and achieve *approximate* conditional validity (Colombo, 2024; Gibbs et al., 2025).

A typical relaxation of exact conditional coverage in earlier work involves group-conditional guarantees (Jung et al., 2023; Ding et al., 2024), which provide coverage guarantees for a predefined set of groups. Another branch of work partitions the covariate space $\mathcal{X}$ into multiple regions and applies CP within each set in the partition (LeRoy & Zhao, 2021; Alaa et al., 2023; Kiyani et al., 2024). However, such partitioning based on the calibration set often leads to overly large prediction regions (Bian & Barber, 2023; Plassier et al., 2024).

An alternative approach weights the empirical cumulative distribution function with a "localizer" function that quantifies the similarity between calibration points and the test sample (Guan, 2023). Although this method improves the localization of predictions, it has significant limitations, especially in high-dimensional covariate spaces.

Finally, several methods focus on the transformation of conformity scores (Han et al., 2022; Dey et al., 2022; Izbicki et al., 2022; Deutschmann et al., 2023; Dheur et al., 2024; Colombo, 2024). These techniques adjust conformity scores to better approximate the conditional coverage. However, they usually require estimating the conditional distribution of conformity scores, which is both computationally intensive and difficult to perform accurately.

In this paper, we propose a novel CP method, *Rectified Conformal Prediction* (RCP), extending normalized nonconformity scores; see, e.g., (Papadopoulos et al., 2008; Papadopoulos & Haralambous, 2011). RCP aims to enhance conditional coverage while preserving exact marginal coverage guarantees. By constructing a new conformity score whose quantile at a given coverage level is independent

of covariates, `RCP` achieves both marginal and improved conditional validity.

A significant benefit of `RCP` is its capacity to generate prediction sets without fully modeling the conditional distribution of conformity scores. Instead, `RCP` concentrates on quantile regression to ensure approximate conditional coverage. The main **contributions** of this work can be summarized as follows.

- We introduce Rectified Conformal Prediction (`RCP`), a new conformal method designed to enhance conditional validity by refining conformity scores (see Sections 3 and 4). The proposed method avoids the need to estimate the full conditional distribution of a multivariate response, relying instead on estimating only the conditional quantile of a univariate conformity score.

- We provide a theoretical lower bound on the conditional coverage of the prediction sets generated by `RCP` (see Section 6). This conditional coverage is explicitly governed by the approximation error in estimating the conditional quantile of the conformity score distribution.

- We evaluate our method on several benchmark datasets and compare it against state-of-the-art alternatives[1] (see Section 7). Our results demonstrate improved performance, particularly in terms of conditional coverage metrics such as worst slab coverage (Romano et al., 2020) and conditional coverage error (Dheur et al., 2024).

## 2. Background

Consider a regression problem that aims to estimate a $d$-dimensional response vector $y \in \mathcal{Y} = \mathbb{R}^d$ based on a feature vector $x \in \mathcal{X} \subseteq \mathbb{R}^p$ to predict. We denote by $\mathrm{P}_{X,Y}$ the joint distribution of $(X, Y)$ over $\mathcal{X} \times \mathcal{Y}$.

Construction of prediction regions for regression problems is often based on distributional regression that focuses on fully characterizing the conditional distribution of a response variable given a covariate (Klein, 2024). This approach improves uncertainty quantification and decision-making (Berger & Smith, 2019). From the conditional predictive distribution, prediction regions can be derived to capture values likely to occur with a given probability. However, these regions rely heavily on the predictive model's quality, and poorly estimated models can result in unreliable predictions. In the following, we present split-conformal prediction (SCP; Papadopoulos et al., 2002), a computationally efficient variant of the conformal prediction framework

that allows generating reliable prediction regions, even when the predictive model is misspecified or inaccurate.

**Split conformal prediction (SCP).** Given a possibly misspecified predictive model $g(x)$, for any input $x \in \mathcal{X}$, SCP (Papadopoulos et al., 2002) generates a prediction set $\mathcal{C}_\alpha(x)$ at a user-specified confidence level $\alpha \in (0, 1)$ with *marginal validity* (Papadopoulos, 2008):

$$\mathbb{P}\big(Y \in \mathcal{C}_\alpha(X)\big) \geq 1 - \alpha. \tag{1}$$

To do so, SCP relies on a *conformity score* function, $V \colon \mathcal{X} \times \mathcal{Y} \to \mathbb{R}$, assigning larger value to worse agreement between $g(X)$ and $Y$. Let $\{(X_k, Y_k)\}_{k=1}^n$ be a calibration set, with $\mathcal{X} \subseteq \mathbb{R}^p$ and $\mathcal{Y} \subseteq \mathbb{R}^d$. SCP generates a prediction set $\mathcal{C}_\alpha(x)$ by computing an empirical quantile of the conformity scores $V(X_k, Y_k), k = 1, \dots, n$:

$$\mathcal{C}_\alpha(x) = \left\{ y \colon V(x, y) \leq Q_{1-\alpha}\left(\textstyle\sum_{k=1}^n \frac{\delta_{V(X_k, Y_k)}}{n+1} + \frac{\delta_\infty}{n+1}\right) \right\},$$

where $\delta_v$ is the Dirac mass at $v$, and $Q_{1-\alpha}(\mathrm{P})$ denotes the $(1-\alpha)$-quantile for any distribution $\mathrm{P}$ on $\mathbb{R}$.

**Towards conditional validity of CP methods.** In many applications, conditional validity is a natural requirement, i.e., for all $x \in \mathcal{X}$,

$$\mathbb{P}\left(Y \in \mathcal{C}_\alpha(X) \mid X = x\right) \geq 1 - \alpha. \tag{2}$$

Conditional coverage (2) is stronger and implies marginal coverage (1). While classical conformal methods provide marginal validity (1), they do not ensure conditional validity.

Let us denote the conditional distribution $\mathrm{P}_{\mathbf{V}|X=x}$ with $\mathbf{V}$ being a shorthand for $V(X, Y)$. The following oracle prediction set

$$\mathcal{C}_\alpha(x) = \left\{ y \in \mathcal{Y} \colon V(x, y) \leq Q_{1-\alpha}(\mathrm{P}_{\mathbf{V}|X=x}) \right\} \tag{3}$$

trivially satisfies conditional coverage (2) by the definition of conditional quantile $Q_{1-\alpha}(\mathrm{P}_{\mathbf{V}|X=x})$. However, exact conditional validity is not achievable within conformal prediction framework (Vovk, 2012; Lei & Wasserman, 2014; Foygel Barber et al., 2021). In what follows we will present a new conformal prediction method that will achieve *approximate* conditional validity while satisfying exact marginal guarantees.

## 3. Rectified Conformal Prediction

The primary objective of our *Rectified Conformal Prediction* (`RCP`) method is to enhance the conditional coverage of any given conformity score while maintaining their exact marginal validity. Expression (3) suggests that one could approximate the $(1-\alpha)$-quantile of the conditional distribution of the scores to construct the prediction set:

$$\tilde{\mathcal{C}}_\alpha(x) = \left\{ y \in \mathcal{Y} \colon V(x, y) \leq \widehat{Q}_{1-\alpha}\big(\mathrm{P}_{\mathbf{V}|X=x}\big) \right\}.$$

---

[1]The code to reproduce main experiments is available at `https://github.com/stat-ml/rcp`

This prediction set provides approximate conditional guarantees that depend on the accuracy of the quantile estimator. However, it fails to ensure exact marginal coverage which is an essential property for conformal prediction methods.

**A motivation for RCP.** Our `RCP` method is specifically designed to achieve both exact conformal marginal validity and approximate conditional coverage. To achieve this, `RCP` first constructs specially transformed (rectified) scores to enhance conditional coverage. To construct the rectified scores, it builds on the key observation that *marginal* and *conditional* coverage coincide precisely when the conditional $(1 - \alpha)$-quantile of the conformity score is independent of the covariates. `RCP` then applies the SCP procedure to these rectified scores, ensuring the classical exact conformal marginal validity.

For any given score $V(x, y)$, referred to as the basic score, `RCP` computes a rectified score $\tilde{V}(x, y)$, which is a transformation of the basic score that satisfies, for $P_X$-a.e. $x \in \mathcal{X}$,

$$Q_{1-\alpha}\big(P_{\tilde{V}(X,Y)}\big) = Q_{1-\alpha}\big(P_{\tilde{V}(X,Y)|X=x}\big). \tag{4}$$

Below we present two examples that show how one can construct the rectified scores satisfying (4).

**Example 1.** Consider the rectified score $\tilde{V}(x, y) = V(x, y)/Q_{1-\alpha}(P_{V|X=x})$, with the assumption that $Q_{1-\alpha}(P_{V|X=x}) > 0$ for any $x \in \mathcal{X}$. We can define the following prediction set, equivalent to (3): $\mathcal{C}_\alpha(x) = \{y \in \mathcal{Y} : \tilde{V}(x, y) \leq 1\}$. This prediction set satisfies conditional coverage. Furthermore, in Appendix B.1, we prove that this rectified score satisfies the equality in (4).

**Example 2.** Consider the rectified score $\tilde{V}(x, y) = V(x, y) - Q_{1-\alpha}(P_{V|X=x})$. The corresponding prediction set, also equivalent to (3), is: $\mathcal{C}_\alpha(x) = \{y \in \mathcal{Y} : \tilde{V}(x, y) \leq 0\}$, and it satisfies conditional coverage. Furthermore, in Appendix B.2, we prove that this rectified score satisfies the equality in (4).

In the following, we generalize over these two basic examples and present a rich family of general score transformations that allow for score rectification.

**`RCP` with general transformations.** Recall that starting from a basic score function $V(x, y)$, we develop a transformed score $\tilde{V}(x, y)$ to achieve conditional validity at a given confidence level $\alpha$. To do so, we introduce a transformation to rectify the basic conformity score $V$.

Consider a parametric family $\{f_t\}_{t \in \mathbb{T}}$ with $(t, v) \in \mathbb{T} \times \mathbb{R} \mapsto f_t(v) \in \mathbb{R}$ and $\mathbb{T} \subseteq \mathbb{R}$. For convenience, we define $\tilde{f}_v(t) = f_t(v)$ and proceed under the following assumption.

**H1.** The function $v \in \mathbb{R} \cup \{\infty\} \mapsto f_t(v)$ is increasing for any $t \in \mathbb{T}$. There exists $\varphi \in \mathbb{R}$ such that $\tilde{f}_\varphi$ is continuous, increasing, and surjective on $\mathbb{R}$.

Under **H**1, we denote by $\tilde{f}_\varphi^{-1}$ the inverse of the function $\tilde{f}_\varphi$, i.e., $\tilde{f}_\varphi^{-1} \circ \tilde{f}_\varphi(t) = t$, for all $t \in \mathbb{T}$. Let $\varphi \in \mathbb{R}$ be such that $\tilde{f}_\varphi$ is invertible (see **H**1). Set

$$V_\varphi(x, y) = \tilde{f}_\varphi^{-1}\big(V(x, y)\big) \tag{5}$$

and denote $\mathbf{V} = V(X, Y)$, and $\mathbf{V}_\varphi = V_\varphi(X, Y)$. We now define the following prediction set

$$\mathcal{C}_\alpha^*(x) = \big\{y \in \mathcal{Y} : V(x, y) \leq f_{\tau_\star(x)}(\varphi)\big\}, \tag{6}$$

where

$$\tau_\star(x) = Q_{1-\alpha}\big(P_{\mathbf{V}_\varphi} \mid X = x\big) = \tilde{f}_\varphi^{-1}\big(Q_{1-\alpha}\big(P_{\mathbf{V}|X=x}\big)\big), \tag{7}$$

i.e., the $(1-\alpha)$ conditional quantile of the transformed score $\mathbf{V}_\varphi$ given $X = x$. We retrieve Example 1 with $f_t(v) = vt$, $\varphi = 1$. In this case $\tilde{f}_1^{-1}(t) = t$ and $V_1(x, y) = V(x, y)$. Similarly, for Example 2, $f_t(v) = v + t$, $\varphi = 0$. In such a case, $\tilde{f}_0^{-1}(t) = t$ and $V_0(x, y) = V(x, y)$.

In the following, we show that the prediction set in (6) satisfies the conditional validity guarantee in (2) and, subsequently, the marginal coverage guarantee in (1). In fact, we can write

$$\mathbb{P}(Y \in \mathcal{C}_\alpha^*(X) \mid X = x) = \mathbb{P}(\mathbf{V} \leq f_{\tau_\star(X)}(\varphi) \mid X = x)$$
$$\overset{(a)}{=} \mathbb{P}(\mathbf{V} \leq \tilde{f}_\varphi(\tau_\star(X)) \mid X = x)$$
$$\overset{(b)}{=} \mathbb{P}(\mathbf{V}_\varphi \leq \tau_\star(X) \mid X = x) \overset{(c)}{\geq} 1 - \alpha,$$

where we have used in (a) that $\tilde{f}_v(t) = f_t(v)$, in (b) that $\tilde{f}_\varphi$ is invertible and the definition of $\mathbf{V}_\varphi$, and in (c) the definition of $\tau_\star(x)$. We may rewrite the prediction set (6) in terms of the rectified score $\tilde{V}_\star(x, y) = f_{\tau_\star(x)}^{-1}\big(V(x, y)\big)$:

$$\mathcal{C}_\alpha^*(x) = \big\{y \in \mathcal{Y} : \tilde{V}_\star(x, y) \leq \varphi\big\}. \tag{8}$$

In Appendix B.4, we establish that the rectified score satisfies (4), more precisely, setting $\tilde{\mathbf{V}}_\star = \tilde{V}_\star(X, Y)$, for all $x \in \mathcal{X}$,

$$\varphi = Q_{1-\alpha}\big(P_{\tilde{\mathbf{V}}_\star|X=x}\big) = Q_{1-\alpha}\big(P_{\tilde{\mathbf{V}}_\star}\big). \tag{9}$$

With the rectified score, conditional and unconditional coverage coincide. However, while the oracle prediction set in (6) provides both conditional and marginal validity, it requires the precise knowledge of the pointwise quantile function $\tau_\star(x)$. In practice, $\tau_\star(x)$ is not known and one must construct an estimate $\hat{\tau}(x)$ using some hold out dataset. Below we discuss the resulting data-driven procedure.

## 4. Implementation of RCP

**The `RCP` algorithm.** Rectified conformal prediction approach, as discussed above, requires a basic conformity

---

**Algorithm 1** The RCP algorithm

**Input:** Calibration dataset $\mathcal{D}$, miscoverage level $\alpha$, conformity score function $V$, transformation function $f_t$, test input $x$.

▷ **Calibration Stage**
Split $\mathcal{D}$ into $\{(X_k, Y_k)\}_{k=1}^{n}$ and $\{(X'_k, Y'_k)\}_{k=1}^{m}$.
$\mathcal{D}_\tau \leftarrow \{(X'_k, V(X'_k, Y'_k))\}_{k=1}^{m}$
$\widehat{q}_{1-\alpha} \leftarrow$ conditional quantile estimate on $\mathcal{D}_\tau$.
$\widehat{\tau} \leftarrow \tilde{f}_\varphi^{-1}(\widehat{q}_{1-\alpha})$
**for** $k = 1$ **to** $n$ **do**
    $\tilde{\mathbf{V}}_k \leftarrow f_{\widehat{\tau}(X'_k)}^{-1}(V(X_k, Y_k))$.
**end for**
$k_\alpha \leftarrow \lceil (1-\alpha)(n+1) \rceil$.
$\tilde{\mathbf{V}}_{(k_\alpha)} \leftarrow k_\alpha$-th smallest value in $\{\tilde{\mathbf{V}}_k\}_{k \in [n]} \cup \{+\infty\}$.
▷ **Test Stage**
    $\mathcal{C}_\alpha(x) \leftarrow \{y \in \mathcal{Y}: f_{\widehat{\tau}(x)}^{-1}(V(x, y)) \leq \tilde{\mathbf{V}}_{(k_\alpha)}\}$.
**Output:** $\mathcal{C}_\alpha(x)$.

---

score function $V$, a transformation function $f_t$, and a calibration dataset of $N = n + m$ points. A critical step in the RCP algorithm is estimating the conditional quantile $\widehat{\tau}(x) \approx Q_{1-\alpha}(\mathrm{P}_{\mathbf{V}_\varphi | X=x})$, which we discuss in detail below. $\widehat{\tau}$ is learned on a separate part of calibration dataset composed of $m$ data points $\{(X'_k, Y'_k): k = 1, \ldots, m\}$. Subsequently, RCP uses SCP with the rectified scores $\tilde{V}(x, y) := f_{\widehat{\tau}(x)}^{-1}(V(x, y))$ instead of the basic scores $V(x, y)$. SCP is applied to the rectified scores computed on the second part of the calibration dataset: $\tilde{\mathbf{V}}_k = \tilde{V}(X_k, Y_k), k = 1, \ldots, n$.

Finally, for a given test input $x$ and miscoverage level $\alpha$, RCP computes the prediction set as

$$\mathcal{C}_\alpha(x) = \left\{ y \in \mathcal{Y}: \tilde{V}(x, y) \leq Q_{1-\alpha}\left(\sum_{k=1}^{n} \frac{\delta_{\tilde{\mathbf{V}}_k}}{n+1} + \frac{\delta_\infty}{n+1}\right)\right\}. \tag{10}$$

The resulting RCP procedure is summarized in Algorithm 1. We show exact marginal validity of RCP and give a bound on its approximate conditional coverage in Section 6 below.

**Estimation of $\tau_\star(x)$.** We present below several methods for estimating $\tau_\star(x)$. Interestingly, even coarse approximations of this conditional quantile can significantly improve conditional coverage; see the discussion in Section 7.

**Quantile regression.** For any $x \in \mathbb{R}^d$, the conditional quantile, denoted by $\tau_\star(x)$, is a minimizer of the expected pinball loss:

$$\tau_\star(x) = \arg\min_\tau \mathbb{E}\left[\rho_{1-\alpha}(V_\varphi(X, Y) - \tau(X))\right], \quad (11)$$

where the minimum is taken over the function $\tau: \mathcal{X} \to \mathbb{R}$ and $\rho_{1-\alpha}$ is the pinball loss (Koenker & Bassett Jr, 1978; Koenker & Hallock, 2001): $\rho_{1-\alpha}(\tau) = (1-\alpha)\tau \mathbb{1}_{\tau>0} - \alpha\tau\mathbb{1}_{\tau \leq 0}$. In practice, the empirical quantile function $\widehat{\tau}$ is

obtained by minimizing the empirical pinball loss:

$$\widehat{\tau} \in \arg\min_{\tau \in \mathcal{C}} \frac{1}{m} \sum_{k=1}^{m} \rho_\alpha(V_\varphi(X'_k, Y'_k) - \tau(X'_k)) + \lambda g(\tau), \tag{12}$$

where $g$ is a penalty function and $\mathcal{C}$ is a class of functions. When $\tau(x) = \theta^\top \Phi(x)$ where $\Phi$ is a feature map, and $g$ is convex, the optimization problem in (12) becomes convex. Theoretical guarantees in this setting, are given, e.g., in (Chen & Wei, 2005; Koenker, 2005).

Non-parametric methods have also been extensively explored, see, e.g., (Chernozhukov & Hansen, 2005; Chernozhukov et al., 2022). Takeuchi et al. (2006) introduced the kernel quantile regression (KQR) framework, formulating quantile regression as minimizing the pinball loss in an RKHS with Tikhonov (squared-norm) regularization. It established some of the first theoretical guarantees for RKHS-based quantile models, deriving finite-sample generalization error bounds using Rademacher complexity; these results were later improved in (Li et al., 2007)

**Local quantile regression.** The local quantile can be obtained by minimizing the empirical weighted expected value of the pinball loss function $\rho_{1-\alpha}$, defined as follows:

$$\widehat{\tau}(x) \in \arg\min_{t \in \mathbb{R}} \left\{ \sum_{k=1}^{m} w_k(x)\rho_{1-\alpha}(V_\varphi(X'_k, Y'_k) - t)\right\}, \tag{13}$$

where $\{w_k(x)\}_{k=1}^{m}$ are positive weights; see (Bhattacharya & Gangopadhyay, 1990). For instance, we can set $w_k(x) = m^{-1}K_{h_X}(\|x - X'_k\|)$, where for $h > 0$, $K_h(\cdot) = h^{-1}K_1(h^{-1}\cdot)$ is a kernel function satisfying $\int K_1(x)\mathrm{d}x = 1$, $\int xK_1(x)\mathrm{d}x = 0$ and $\int x^2K_1(x)\mathrm{d}x < \infty$; $h_X$, the kernel bandwidth is tuned to balance bias and variance. With appropriate adaptive choice of $h(x)$, this approach can be shown to be asymptotically minimax over Hölder balls; see (Bhattacharya & Gangopadhyay, 1990; Spokoiny et al., 2013; Reiß et al., 2009). More recently, Shen et al. (2024) introduced a penalized non-parametric approach to estimating the quantile regression process (QRP) using deep neural networks with rectifier quadratic unit (ReQU) activations. Shen et al. (2024) derives upper bounds on the mean-squared error for quantile regression using deep ReQU networks, depending only on the approximation error and network. The bounds are shown to be tight for broad function classes (e.g., Hölder compositions, Besov spaces), implying that ReQU neural networks achieve minimax-optimal convergence rates for conditional quantile estimation. Notably, the theory requires minimal assumptions and holds even for heavy-tailed error distributions.

## 5. Related Work

It is well known that obtaining exact conditional coverage for all possible inputs within the conformal prediction framework is impossible without making distributional assumptions (Foygel Barber et al., 2021). However, the literature has proposed various relaxations of exact conditional coverage, focusing on different notions of approximate conditional coverage.

A first class of methods involves group-conditional guarantees (Jung et al., 2023; Ding et al., 2024), which provide coverage guarantees for a predefined set of groups. Another class partitions the covariate space into multiple regions and applies classical conformal prediction within each region (LeRoy & Zhao, 2021; Alaa et al., 2023; Kiyani et al., 2024). The significant limitation of these methods lies in the need to specify the groups or regions in advance.

Other conformal methods aim to approximate conditional coverage by leveraging uncertainty estimates from the base predictor. When $d = 1$, Conformalized Quantile Regression (CQR; Romano et al., 2019) suggests constructing a conformalized prediction interval $\mathcal{C}_\alpha(x)$ by leveraging two quantile estimates of $Y \mid X = x$, denoted as $\hat{q}_{\alpha/2}(x)$ and $\hat{q}_{1-\alpha/2}(x)$. This approach yields prediction intervals that adapt to heteroscedasticity (Kivaranovic et al., 2020). By considering a version of CQR by Sesia & Candès (2020) whose conformity score is positive, we can draw a connection with RCP. The conformity score is

$$V(x, y) = |y - \mu_\alpha(x)| / \delta_\alpha(x),$$

with $\mu_\alpha(x) = (\hat{q}_{1-\alpha/2}(x) + \hat{q}_{\alpha/2}(x))/2$ and $\delta_\alpha(x) = \hat{q}_{1-\alpha/2}(x) - \hat{q}_{\alpha/2}(x)$. Applying RCP with $f_t(v) = tv$ yields the following scaled transformed conformity scores:

$$\tilde{V}(x, y) = |y - \mu_\alpha(x)| / \hat{\tau}(x),$$

where $\hat{\tau}(x)$ is an estimator of the conditional $(1 - \alpha)$-quantile of $|Y - \mu_\alpha(x)|$ given $X = x$. Thus, this particular variant of RCP closely resembles the CQR approach but uses a different quantile estimate.

In the context of multivariate prediction sets, given a predictor $\mu(\cdot)$, a natural choice for the conformity score is $V_\infty(x, y) = \|y - \mu(x)\|_\infty$, where $\|u\|_\infty = \max_{1 \le t \le d}(|u_i|)$ (Diquigiovanni et al., 2024). This conformity score measures the prediction error associated with the predictor $\mu$ (Nouretdinov et al., 2001; Vovk et al., 2005; 2009). Setting $f_t(v) = tv$ and $\varphi = 1$, the rectified conformity scores are given by $\tilde{V}(x, y) = V(x, y)/\hat{\tau}(x)$ where $\hat{\tau}(x) \approx Q_{1-\alpha}(\mathrm{P}_{\mathbf{V}_\infty \mid X = x})$, with $\mathbf{V}_\infty = V_\infty(X, Y)$. Thus, RCP is similar to the approach proposed in (Lei et al., 2018), but with a different choice of scaling function.

Methods utilizing conditional density estimation have been proposed to produce conformal prediction intervals that adapt to skewed data (Sesia & Romano, 2021), to minimize the average volume (Sadinle et al., 2019, denoted DCP in our paper) or to define more flexible highest-density regions (Izbicki et al., 2022; Plassier et al., 2025). Probabilistic conformal prediction (PCP; Wang et al., 2023) bypasses density estimation by constructing prediction sets as unions of balls centered on samples from a generative model. All these methods are either tailored to handle the scalar response ($d = 1$) or require an accurate conditional distribution estimate which might be hard to obtain in practical scenarios.

Guan (2023) introduces a localized conformal prediction framework that adapts to data heterogeneity by weighting calibration points based on their similarity to the test sample. To do so, kernel-based localizers assign greater importance to nearby points, tailoring prediction intervals to local data patterns. Amoukou & Brunel (2023) extend Guan's approach by replacing kernels with quantile regression forest estimators for improved performance. Although effective, these methods face challenges in high-dimensional or mixed-variable settings.

Several methods aim to transform conformity scores to improve approximate conditional coverage. For example, Johansson et al. (2021), following earlier works by Papadopoulos & Haralambous (2011); Johansson et al. (2014); Lei et al. (2018), investigate *normalized conformity scores* (NCF), which enhance standard conformal prediction by adjusting prediction set according to instance difficulty. NCF can be represented within our framework through a specific choice of the function $f_\tau(v) = v/(\tau + \beta)$, where $\beta$-values will put a greater emphasis on the difficulty estimation. Notably, the estimation approach employed in these papers uses least-squares regression on residuals, in contrast to the quantile regression approach adopted in RCP, which is essential to satisfy (4). Han et al. (2022) presents an approach that uses kernel density estimation to approximate the conditional distribution. Similarly, Deutschmann et al. (2023) rescales the conformity scores based on an estimate of the local score distribution using the jackknife+ technique. However, these methods generally rely on estimating the conditional distribution of conformity scores, which is challenging in practice. Dewolf et al. (2025) studies conditional validity of normalized conformal predictors in oracle setting, i.e., when the optimal normalization is known.

Recent work by Colombo (2024) suggests to transform the conformity score employing a normalizing flow: $\tilde{V}(x, y) = b(V(x, y), x)$. The normalizing flow is trained to map the joint distribution $\mathrm{P}_{V, X}$ of the conformity score and attributes into a product distribution, $\mathrm{P}_{\tilde{\mathbf{V}}} \otimes \mathrm{P}_X$, where $\mathrm{P}_{\tilde{\mathbf{V}}}$ is an arbitrary univariate distribution. Notably, this condition is stricter than the conditional coverage criterion (4), as it enforces $\mathrm{P}_{\tilde{\mathbf{V}} \mid X = x} = \mathrm{P}_{\tilde{\mathbf{V}}}$ for almost every $x$ under $\mathrm{P}_X$. Con-

sequently, learning such a transformation typically necessitates a larger sample size; see Section 7.

One method (Xie et al., 2024) proposes to use a cross-validated boosting procedure to learn a new score function to be used in split-conformal prediction. The authors consider a specific family of possible score functions and corresponding loss functions tailored either to deviation from conditional coverage or interval length. This method has several limitations compared to our approach: limited set of score functions, tailored to one-dimensional targets, requires access to the train set, and high computation cost.

RCP method shares some similarities with that of Gibbs et al. (2025), which also performs a quantile regression of the conformity score with respect to the attribute $X$. There are two essential differences: firstly, Gibbs et al. (2025) work directly with the conformity score $V$, whereas we regress on a transformed score $V_\varphi$. Secondly, the manner in which the quantile regression result is used differs significantly. RCP uses the quantile estimator to define the rectified score $\tilde{V}$, to which the standard CP procedure, while Gibbs et al. (2025) propose a considerably more complex procedure; see Section 7.

Finally, various recalibration methods have been proposed to improve marginal coverage (Dheur & Taieb, 2023) or conditional coverage (Dey et al., 2022). While these methods can also be interpreted within the conformal prediction framework (Marx et al., 2022; Dheur & Taieb, 2024), they often require modifications to the training procedure, making them less broadly applicable than purely conformal methods.

## 6. Theoretical Guarantees

In this section, we study the marginal and conditional validity of the predictive set $\mathcal{C}_\alpha(x)$ defined in (10). Due to space constraints, we present simplified versions of the results. Full statements and rigorous proofs can be found in the supplement materials. Many of the results hold independently of the specific method used to construct the conditional quantile estimator $\hat{\tau}(x)$. The only assumption we impose is minimal

**H2.** For any $x \in \mathcal{X}$, we have $\hat{\tau}(x) \in \mathbb{T}$.

The following theorem establishes the standard conformal guarantee. We stress that for this statement, the definition of $\hat{\tau}(x)$ is not essential. The result is valid for any function $\tau(x)$, and the proof follows directly from classical arguments demonstrating the validity of split-conformal method.

**Theorem 1.** *Assume **H1**-**H2** hold and suppose the rectified conformity scores $\{\tilde{\mathbf{V}}_k\}_{k=1}^{n+1}$ are almost surely distinct. Then,*

*for any $\alpha \in (0, 1)$, it follows*

$$1 - \alpha \leq \mathbb{P}\left(Y_{n+1} \in \mathcal{C}_\alpha(X_{n+1})\right) < 1 - \alpha + \frac{1}{n+1}.$$

The proof is postponed to Appendix B.3. We will now examine the conditional validity of the prediction set. To do so, we will explore the relationship between the conditional coverage of $\mathcal{C}_\alpha(x)$ and the accuracy of the conditional quantile estimator $\hat{\tau}(x)$. To simplify the statements, we assume that the distribution of $P_{\mathbb{V}_\varphi | X = x}$, where $\mathbf{V}_\varphi = V_\varphi(X, Y)$ is continuous. Define

$$\epsilon_\tau(x) = \mathbb{P}\left(V_\varphi(X, Y) \leq \tau(x) \mid X = x\right) - 1 + \alpha. \quad (14)$$

The function $\epsilon_\tau$ represents the deviation between the current confidence level and the desired level $1 - \alpha$. Define the conditional pinball loss

$$\mathcal{L}_x(\tau) = \mathbb{E}\left[\rho_{1-\alpha}\left(V_\varphi(X, Y) - \tau(X)\right) \mid X = x\right]. \quad (15)$$

It is shown in Theorem 6 (see Appendix B.6) that, under weak technical conditions, $\epsilon_\tau$ satisfies the following property: for all $x \in \mathcal{X}$,

$$|\epsilon_\tau(x)| \leq \sqrt{2 \times \{\mathcal{L}_x(\tau(x)) - \mathcal{L}_x(\tau_\star(x))\}},$$

where $\tau_\star(x)$ is defined in (7). The previous equation bounds $\epsilon_\tau(x)$ as a function of the quantile estimate $\tau(x)$. If $\tau(x)$ is close to the minimizer of the loss function $\mathcal{L}_x$ (as defined in (15)), then $\epsilon_\tau(x)$ is expected to approach zero.

The c.d.f function of the rectified conformity score is defined as $F_{\tilde{V}} = \mathbb{P}(\tilde{V}(X, Y) \leq \cdot)$. We denote its conditional version by $F_{\tilde{V}|X=x} = \mathbb{P}(\tilde{V}(x, Y) \leq \cdot \mid X = x)$.

**Theorem 2.** *Assume that **H1**-**H2** and $F_{\tilde{V}}$ is continuous and that, for any $x \in \mathcal{X}$, $F_{\tilde{V}|X=x} \circ F_{\tilde{V}}^{-1}$ is L-Lipschitz. Then, for any $\alpha \in [\{n+1\}^{-1}, 1)$ it holds*

$$\mathbb{P}\left(Y_{n+1} \in \mathcal{C}_\alpha(X_{n+1}) \mid X_{n+1} = x\right) \geq 1 - \alpha \\ + \epsilon_{\hat{\tau}}(x) - \alpha \mathrm{L} \times [F_{\tilde{V}}(\varphi)]^{n+1}. \quad (16)$$

The proof is postponed to Appendix B.5. According to Theorem 2, the conditional validity of the prediction set $\mathcal{C}_\alpha(x)$ directly depends on the accuracy of the quantile estimate $\hat{\tau}(x)$. If $\hat{\tau}(x)$ closely approximates the conditional quantile $Q_{1-\alpha}\left(P_{V_\varphi(x,Y)|X=x}\right)$, then (16) ensures that conditional coverage is approximately achieved.

**Local quantile regression.** We will now explicitly control $\epsilon_{\hat{\tau}}(x)$ when the estimate $\hat{\tau}(x)$ is obtained using the local quantile regression method outlined in (13). For any $x \in \mathbb{R}^d$, we define $C_{h_X}(x)$ as $C_{h_X}(x) = \mathbb{E}[K_{h_X}(\|x - X\|)]$.

**H3.** There exists $\mathrm{M} \geq 0$, such that for all $v \in \mathbb{R}$, $\tilde{x} \mapsto F_{V_\varphi(\tilde{x}, Y)|X=\tilde{x}}(v)$ is M-Lipschitz. Moreover, $t \in \mathbb{R}_+ \mapsto K_{h_X}(t)$ is non-increasing.

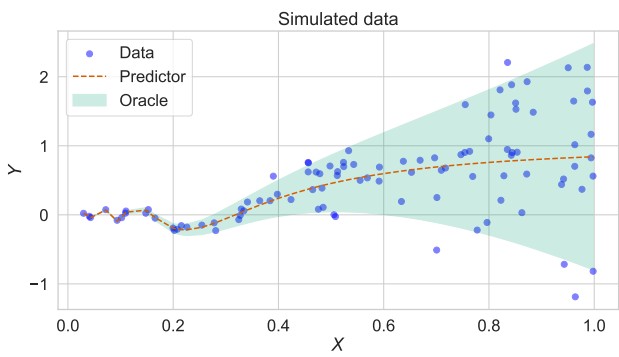

Figure 1: Oracle data distribution, sample data and predictor for the toy dataset.

**Proposition 1.** *Assume **H3** holds. With probability at most* $m^{-1} \times \{1 + 4C_{h_X}(x)^{-1} \operatorname{Var}[K_{h_X}(\|x - X\|)]\}$, *it holds*

$$|\epsilon_{\widehat{\tau}}(x)| \geq C_{h_X}(x)^{-1} \sqrt{\frac{K_1(0) \log m}{h_X m}}$$
$$+ 4C_{h_X}(x)^{-1} \sup_{0 \leq t \leq 1} \{t K_{h_X}(\mathrm{M}^{-1} t)\}.$$

The proof is postponed to Appendix B.7. Proposition 1 highlights the trade-off associated with the bandwidth parameter $h_X$. Ideally, we would like to choose $h_X \ll 1$ to minimize $\sup_{0 \leq t \leq 1} \{t K_{h_X}(\mathrm{L}^{-1} t)\}$. However, this results in an increase of $\sqrt{\frac{\log m}{m h_X}}$. Consequently, there exists an optimal bandwidth parameter $h_X$ that depends on both the number of available data points $m$ and the regularity of the conditional cumulative distribution function $x \mapsto F_{V_\varphi(x,Y)|X=x}(v)$.

Finally, for the optimal choice of bandwidth $h_X$ one can prove the asymptotic validity of RCP:

$$\mathbb{P}(Y \in \mathcal{C}_\alpha(X) \mid X) \to 1 - \alpha, \quad n, m \to \infty. \tag{17}$$

The exact formulation and its proof are given in Appendix B.8.

# 7. Experiments

## 7.1. Toy example

Let us consider the following data-generating process:

$$X \sim \mathrm{Beta}(1.2, 0.8), \quad Y \mid X = x \sim \mathcal{N}(\mu(x), x^4).$$

where $\mu(x) = x \sin(x)$. Figure 1 shows a realization with $n = 100$ data points. Our goal is to investigate the influence of the quality of the $(1 - \alpha)$-quantile estimate $\widehat{\tau}$ on performance.

We set $\alpha = 0.1$ and consider the conformity score $V(x, y) = |y - \mu(x)|$. In this case, the $(1 - \alpha)$-quantile of

| $\omega$ | 0 | $1/3$ | $2/3$ | 1 |
|---|---|---|---|---|
| COVERAGE | $90 \pm 01$ | $84 \pm 01$ | $75 \pm 03$ | $59 \pm 07$ |

Table 1: Local coverage on the adversarially selected 10% of the data, $\omega$ corresponds to the level of contamination of the score quantile estimate.

$V(x, Y) \mid X = x$ is known and we denote it by $Q_{1-\alpha}(x)$. Given $\omega \in [0, 1]$, we set $\widehat{\tau}(x) \sim (1 - \omega) Q_{1-\alpha}(x) + \omega \epsilon(x)$, where we consider $\epsilon(x) \sim \mathcal{N}(0, x^4)$. We perform 1000 experiments and report the 10% lower value of $x \in [0, 1] \mapsto \mathbb{P}(Y_{n+1} \in \mathcal{C}_\alpha(x) \mid X = x)$; the results can be found in Table 1. If $\omega = 0$, $\widehat{\tau}(x)$ corresponds to the true $(1 - \alpha)$-quantile. In this case, our method is conditionally valid, as Theorem 2 shows. However, while all settings of $\omega$ yield marginally valid prediction sets, the conditional coverage decreases as the quantile estimate $\widehat{\tau}(x)$ deteriorates.

## 7.2. Real-world experiment

We use publicly available regression datasets which are also considered in (Tsoumakas et al., 2011; Feldman et al., 2023; Wang et al., 2023) and only keep datasets with at least 2 outputs and 2000 total instances. The characteristics of the datasets are summarized in Appendix C.

**Base predictors.** We consider two base predictors, both parameterized by a fully connected neural network with three layers of 100 units and ReLU activations.

The *mean predictor* estimates the mean $\hat{\mu}_i(x)$ of the distribution for each dimension $i \in [d]$ given $x \in \mathcal{X}$. Since it only provides a point estimate, it does not capture uncertainty.

The *mixture predictor* models a mixture of $K$ Gaussians, enabling it to represent multimodal distributions. Given $x \in \mathcal{X}$, the model outputs $z(x) \in \mathbb{R}^K$ (logits for mixture weights), $\mu(x) \in \mathbb{R}^{K \times d}$ (mean vectors), and $L(x) \in \mathbb{R}^{K \times d \times d}$ (lower triangular Cholesky factors). The mixture weights $\pi(x) \in \mathbb{R}^K$ are obtained by applying the softmax function to $z(x)$, and the covariance matrices $\Sigma(x) \in \mathbb{R}^{K \times d \times d}$ are computed as $\Sigma_k(x) = L_k(x) L_k(x)^\top$. The conditional density at $y \in \mathcal{Y}$, given $x \in \mathcal{X}$, is:

$$\hat{p}(y \mid x) = \sum_{k=1}^{K} \pi_k(x) \cdot \mathcal{N}(y \mid \mu_k(x), \Sigma_k(x)),$$

where $\mathcal{N}(y \mid \mu_k(x), \Sigma_k(x))$ is a Gaussian density with mean $\mu_k(x)$ and covariance matrix $\Sigma_k(x)$.

**Methods.** We compare RCP with four split-conformal prediction methods from the literature: ResCP (Diquigiovanni et al., 2024), PCP (Wang et al., 2023), DCP (Sadinle et al., 2019), and SLCP (Han et al., 2022). ResCP uses residuals as conformity scores. To handle multi-dimensional outputs,

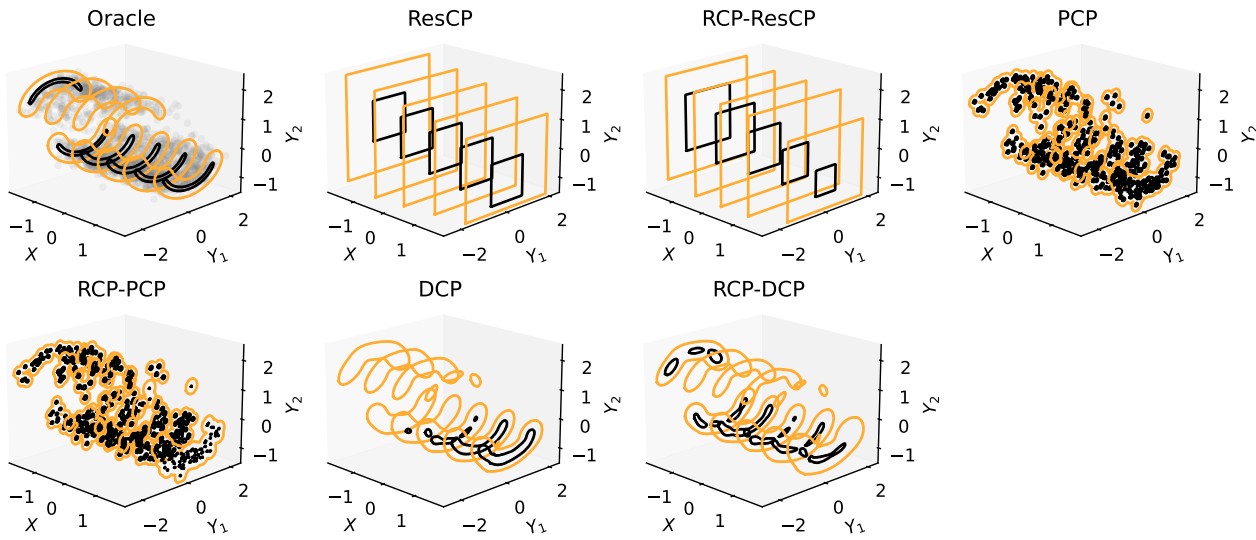

Figure 2: Examples of prediction sets on synthetic dataset where the output has a bivariate and bimodal distribution.

we follow (Diquigiovanni et al., 2024) and define the conformity score as the $l^\infty$ norm of the residuals across dimensions, i.e., $V(x, y) = \max_{i \in [d]} |\hat{\mu}_i(x) - y_i|$. PCP constructs the prediction set as a union of balls, while DCP defines the prediction set by thresholding the density. ResCP is compatible with the *mean predictor*, whereas PCP and DCP are compatible with the *mixture predictor*. Finally, SLCP, like RCP, is compatible with any conformity score and base predictor. For RCP, we compute an estimate $\hat{\tau}(x)$ (see Section 4) using quantile regression with a fully connected neural network composed of 3 layers with 100 units.

**Visualization on a synthetic dataset.** Figure 2 illustrates example prediction sets for different methods. The orange and black contour lines represent confidence levels of $\alpha = 0.1$ and $\alpha = 0.8$, respectively. The first panel shows the highest density regions of the oracle distribution, while the subsequent panels display prediction regions obtained by different methods, both before and after applying RCP. We can see that combining RCP with ResCP, PCP, or DCP results in prediction sets that more closely align with those of the oracle distribution.

**Experimental setup.** We reserve 2048 points for calibration. The remaining data is split between 70% for training and 30% for testing. The base predictor is trained on the training set, while the baseline conformal methods use the full calibration set to construct prediction sets for the test points. In RCP, the calibration set is further divided into two parts: one for estimating $\hat{\tau}(x)$ and the other as the proper calibration set for obtaining intervals. This ensures that all methods use the same number of points for uncertainty estimation. When not specified, we used the adjustment $f_t(v) = t + v$. Additional details on implementation and

hyperparameter tuning are provided in Appendix C.

**Evaluation metrics.** To evaluate conditional coverage, we use *worst-slab coverage* (WSC, Cauchois et al., 2020; Romano et al., 2020) with $\delta = 0.2$ and the *conditional coverage error*, computed over a partition of $\mathcal{X}$, following Dheur et al. (2025). To evaluate sharpness, we also report the median of the logarithm of the prediction set volume, scaled by the dimension $d$.

**Main results.** Figure 3 presents the worst-slab coverage and volume for different conformity scores, both with and without RCP. Similarly, Figure 4 compares worst-slab coverage between SLCP and RCP. Additional results, including conditional coverage error and marginal coverage, are provided in Appendix A.1.

In the left panel of Figure 3, we observe that ResCP, PCP, and DCP fail to reach the nominal level of conditional coverage for most datasets. In contrast, all variants of RCP significantly improve coverage across all datasets. Similarly, Figure 4 shows that RCP often achieves better conditional coverage than SLCP, particularly on larger datasets. Figure 5 in Appendix A.1 confirms these findings with the conditional coverage error. Finally, as expected, all methods achieve marginal coverage.

In the right panel of Figure 3, we observe that RCP improves the median prediction set volume compared to non-RCP variants in addition of improving conditional coverage.

Finally, Appendix A.6 compares average and median volumes of prediction sets produced by direct conformal methods and their RCP counterparts. Direct methods obtain a smaller average volume while RCP obtains a smaller median

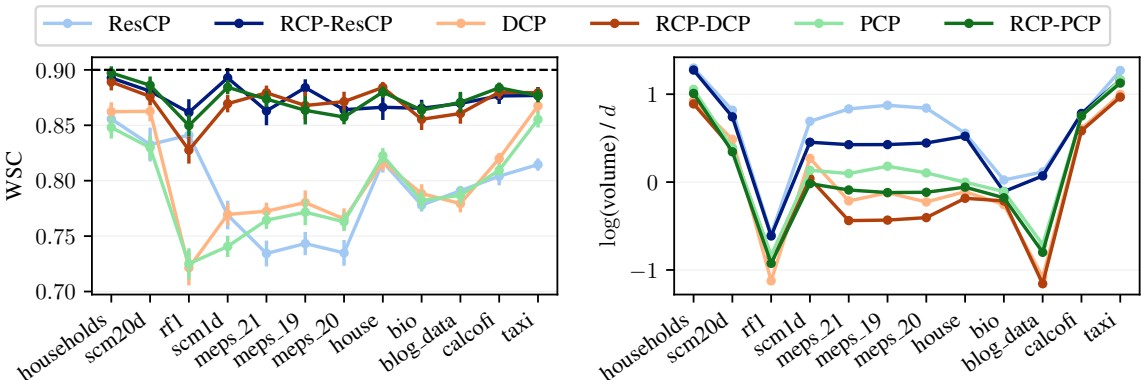

Figure 3: Worst-slab coverage and volume for three conformal methods and their RCP counterparts, on datasets sorted by total size.

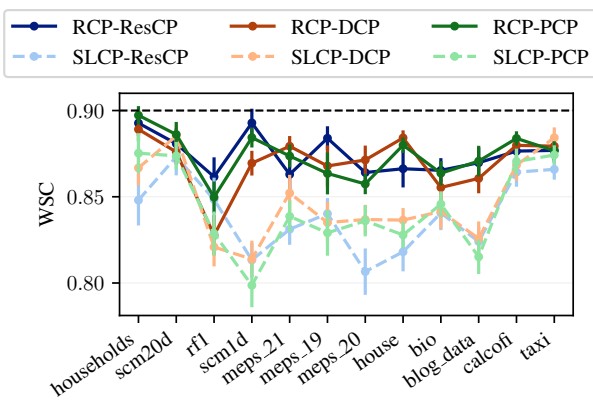

Figure 4: Worst-slab coverage for RCP and SLCP in combination with different conformity scores, on datasets sorted by total size.

volume.

**Additional experiments.** We complement main results with multiple experiments aiming at studying variations of the proposed method and comparing it with some additional competitors.

Appendix A.2 discusses the estimation of $\widehat{\tau}(x)$ using either a neural network or local quantile regression for which we have bounds the conditional coverage. On most datasets, the neural network slightly outperforms local quantile regression, which is expected due to its flexibility. Appendices A.3 and A.4 discuss the choice of adjustment function. For certain adjustment functions, the domain of the scores $v = V(x, y)$ must be restricted to a subset of $\mathbb{R}$ to satisfy **H**2. Notably, $f_t(v) = tv$ requires $v > 0$, $f_t(v) = \exp(tv)$ requires require $v > 1$.

Appendix A.5 directly compares the proposed method with

CQR, showing that CQR already obtains a competitive conditional coverage but is outperformed by RCP-DCP in average volume. Appendix A.7 presents an additional study comparing RCP with CP and CQR methods, that we adapted to multidimensional target setting. For these experiments, the model predicts parameters of a multivariate normal distribution and we use the score based on the corresponding Mahalanobis distance. We demonstrate that RCP improves conditional coverage over classic CP and also benefits from the custom score to outperform CQR.

Appendix A.8 considers an approach to improve data efficiency. Instead of dividing the calibration dataset $\mathcal{D}$ into two parts to estimate $\widehat{\tau}$, we compute out-of-sample conformity scores on the training dataset $\mathcal{D}_{\text{train}}$ using $K$-fold cross-validation. This results in improved conditional coverage at the cost of training $K$ additional models.

Appendix A.9 provides an additional comparison with Conditional Prediction with Conditional Guarantees (CPCG; Gibbs et al. (2025)). CPCG obtains a competitive conditional coverage but is 200-100000 times slower than RCP overall, limiting its applicability.

## 8. Conclusion

We present a new approach to improve the conditional coverage of the conformal prediction set while preserving marginal convergence. Our method constructs prediction sets by adjusting the conformity scores using an appropriately defined conditional quantile, allowing RCP to automatically adapt the prediction sets against heteroscedasticity. Our theoretical analysis supports that this approach produces approximately conditionally valid prediction sets; furthermore, the theory provides lower bounds on the conditional coverage, which explicitly depends on the distribution of the conditional quantile estimator $\widehat{\tau}(x)$.

## Acknowledgments

V.P. carried out this study during his PhD under the supervision of E.M., and subsequently as a research engineer at the Centre Lagrange in Paris. Part of E.M.'s work was also conducted under the auspices of the Centre Lagrange. E.M. was all partially funded by the European Union (ERC-2022-SyG, 101071601). Views and opinions expressed are however those of the author(s) only and do not necessarily reflect those of the European Union or the European Research Council Executive Agency. Neither the European Union nor the granting authority can be held responsible for them.

## Impact Statement

This work contributes to the broader effort to improve the interpretability and statistical reliability of machine learning algorithms. Prediction intervals with exact marginal and approximate conditional coverage offer a useful tool for conveying uncertainty regarding the accuracy of ML models, which is essential for increasing their fairness, and fostering wider acceptance.

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

# A. Additional Experiments

## A.1. Additional results on marginal coverage and conditional coverage error

Figure 5 extends the results of Figure 3 by displaying additionally the marginal coverage and conditional coverage error. As expected, all methods obtain a correct marginal coverage. Furthermore, the methods with the best worst slab coverage (closest to $1 - \alpha$) also obtain a small conditional coverage error, supporting our conclusions in Section 7.

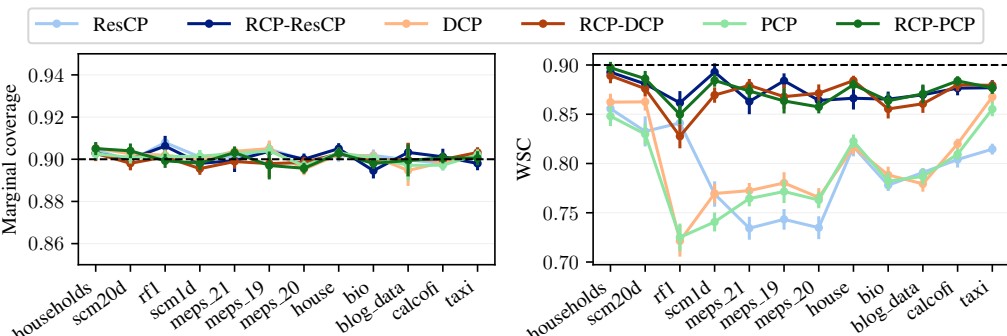

Figure 5: Marginal coverage and conditional coverage error for three conformal methods and their `RCP` counterparts, on datasets sorted by total size.

## A.2. Estimation of conditional quantile function

Figure 6 compares two ways of estimating $\widehat{\tau}$ (see Section 4). $\text{RCP}_{\text{MLP}}$ corresponds to quantile regression based on a neural network as in Section 7, while $\text{RCP}_{\text{local}}$ corresponds to local quantile regression. On many datasets, the more flexible $\text{RCP}_{\text{MLP}}$ is able to obtain better conditional coverage. However, local quantile regression has theoretical guarantees on its conditional coverage (see Section 6).

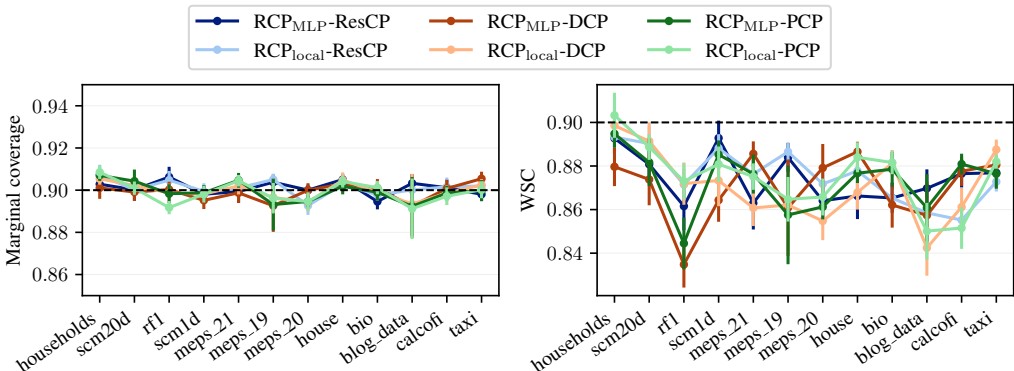

Figure 6: Marginal coverage and conditional coverage error for two types of quantile estimators in combination with different conformal methods, on datasets sorted by total size.

### A.3. Choice of adjustment function

Figure 7 compares RCP with difference ($-$) and linear ($*$) adjustments when combined with the DCP method. Since RCP with any adjustment function adheres to the SCP framework, marginal coverage is guaranteed, as shown in Panel 1.

The conformity score for DCP is defined as $V(x, y) = -\log \hat{p}(y \mid x)$, which can take negative values, implying that $\mathbb{T} = \mathbb{R}$. However, the linear adjustment requires $\mathbb{T} \subseteq \mathbb{R}_+^*$, violating $\mathbf{H}1$ and resulting in a failure to approximate conditional coverage accurately. This issue is evident in Panel 2. In contrast, the difference adjustment does not impose such a restriction.

Panel 3 compares PCP and ResCP when used with difference and linear adjustments. Since the conformity scores for these methods are always positive, i.e., $\mathbb{T} = \mathbb{R}_+^*$, both adjustment methods satisfy $\mathbf{H}1$. In general, we observe no significant differences between the two adjustment methods.

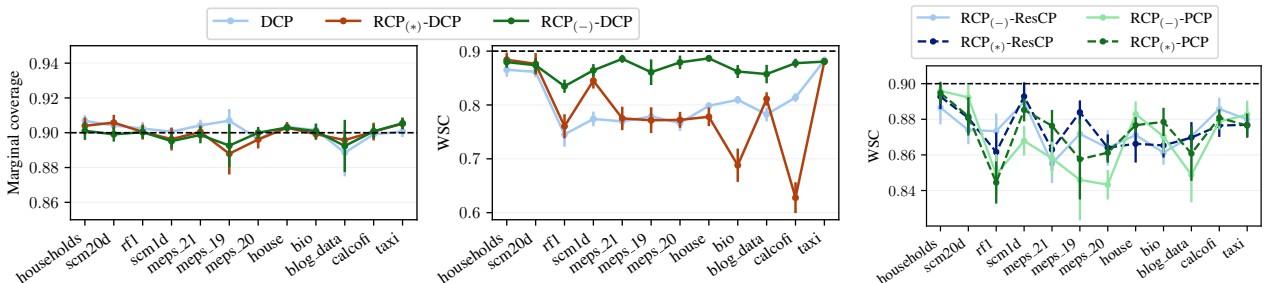

Figure 7: Marginal coverage and conditional coverage error obtained for two types of adjustments.

### A.4. Additional adjustment functions

We consider two additional adjustments functions, namely $f_t(v) = \exp(t + v)$, denoted $\exp -$, and $f_t(v) = \exp(tv)$, denoted $\exp *$. To apply these custom adjustment functions we need to ensure that the conditions $\mathbf{H}1$ and $\mathbf{H}2$ are satisfied. For the first function we have: $\tilde{f}_\varphi^{-1}(v) = (\ln v) - \varphi \in \mathbb{T}$ and $\varphi = 0$. Then $\tilde{f}_\varphi^{-1}(v) > 0 \Rightarrow \ln v > 0 \Rightarrow v > 1$. For the second function we can take $\varphi = 1$ and by similar argument we arrive at the same requirement $v > 1$. In practice, conformity scores are usually non-negative as is the case with PCP and residual scores that we consider here, and we can always add a constant 1 to satisfy this requirement.

Figures 8 and 9 show the marginal coverage and conditional coverage error obtained with these adjustment functions.

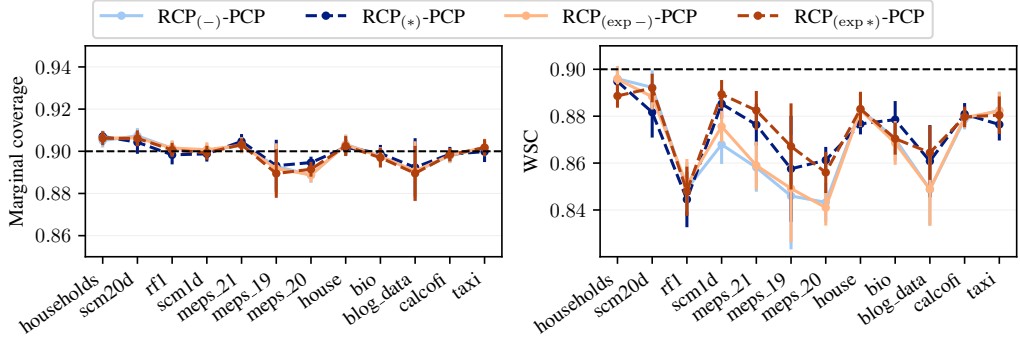

Figure 8: Marginal coverage and conditional coverage error for two additional types of adjustments combined with the method PCP.

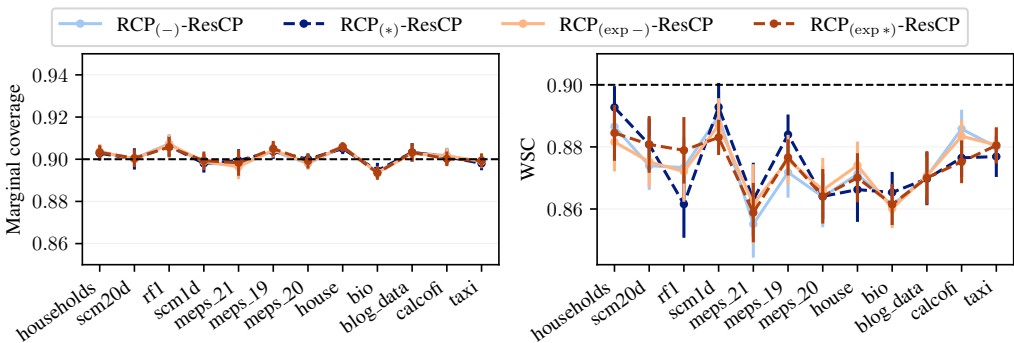

Figure 9: Marginal coverage and conditional coverage error for two additional types of adjustments combined with the method ResCP.

### A.5. Direct comparison with CQR

Here we present a direct comparison of RCP with Conformalized Quantile Regression (CQR; Romano et al. (2019)). We use the same underlying neural network architectures for the models as in our main experiment. Similarly to ResCP, to handle multi-dimensional outputs, we follow (Diquigiovanni et al., 2024) and define the conformity score of CQR as the $l^\infty$ norm of the CQR conformity scores across dimensions. Specifically, we compare CQR to DCP and its RCP-DCP counterpart, which achieves the best median volume overall.

Figure 10 shows that CQR matches the conditional coverage of RCP-DCP. However, it produces larger median prediction sets due to less flexible shapes.

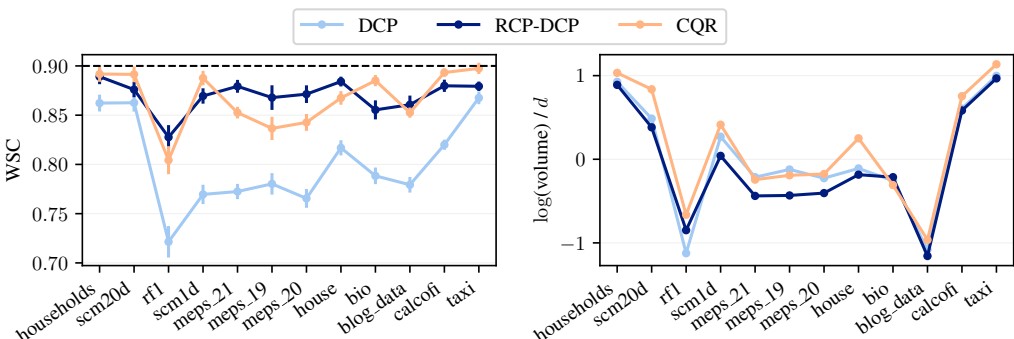

Figure 10: Worst slab coverage and (logarithm) median prediction set volume (scaled by $d$).

## A.6. Comparison of prediction set volumes

Table 2 shows the average volume obtained by the methods compared in Section 7. Non-RCP methods obtain a smaller average volume across all datasets. The larger average volume of RCP is explained by the larger regions produced for instances with larger uncertainty.

Table 2: Mean prediction set volume per dataset.

| dataset | PCP | RCP-PCP | DCP | RCP-DCP | ResCP | RCP-ResCP |
|---|---|---|---|---|---|---|
| households | 88.3 | 1.33e+02 | **47.4** | 1.02e+02 | 1.81e+02 | 4.51e+02 |
| scm20d | **4.26e+05** | 7.92e+06 | 1.11e+06 | 3.95e+07 | 5.22e+05 | 7.15e+12 |
| rf1 | 0.0274 | 0.190 | **0.000562** | 4.35e+04 | 0.0276 | 7.94e+08 |
| scm1d | **1.92e+04** | 1.30e+08 | 2.30e+04 | 1.67e+08 | 6.27e+04 | 2.04e+15 |
| meps_21 | 1.65 | 10.0 | **0.746** | 6.07 | 5.35 | 8.32e+12 |
| meps_19 | 90.0 | 3.27e+04 | **3.64** | 3.27e+04 | 5.56 | 3.56e+22 |
| meps_20 | 1.68 | 5.50 | **0.761** | 6.20 | 5.38 | 6.27e+13 |
| house | 0.676 | 0.936 | **0.519** | 0.751 | 2.92 | 3.88 |
| bio | 0.579 | 1.12 | **0.414** | 0.645 | 1.05 | 1.16 |
| blog_data | 0.459 | 8.45e+02 | **0.143** | 6.37e+02 | 1.26 | 6.79e+21 |
| calcofi | 3.47 | 4.12 | **2.45** | 3.06 | 4.53 | 4.47 |
| taxi | 9.21 | 9.63 | **5.69** | 6.40 | 12.4 | 12.8 |

In contrast, Table 3 shows that RCP obtains smaller regions across most datasets when comparing the median volume, avoiding outliers.

Table 3: Median prediction set volume per dataset.

| dataset | PCP | RCP-PCP | DCP | RCP-DCP | ResCP | RCP-ResCP |
|---|---|---|---|---|---|---|
| households | 67.4 | 56.5 | 39.2 | **32.1** | 1.81e+02 | 1.67e+02 |
| scm20d | 3.11e+04 | **1.42e+04** | 7.03e+05 | 7.74e+04 | 5.22e+05 | 2.00e+05 |
| rf1 | 0.0110 | 0.00525 | **0.000583** | 33.1 | 0.0276 | 0.0231 |
| scm1d | 1.16e+02 | **2.05** | 1.01e+04 | 25.7 | 6.27e+04 | 1.70e+03 |
| meps_21 | 1.04 | 0.704 | 0.433 | **0.227** | 5.35 | 2.42 |
| meps_19 | 3.85 | 0.754 | 2.10 | **0.303** | 5.56 | 2.54 |
| meps_20 | 1.05 | 0.616 | 0.416 | **0.254** | 5.38 | 2.51 |
| house | 0.596 | 0.519 | 0.471 | **0.386** | 2.92 | 2.67 |
| bio | 0.507 | 0.435 | 0.374 | **0.344** | 1.05 | 0.829 |
| blog_data | 0.229 | 0.209 | 0.0869 | **0.0597** | 1.26 | 1.16 |
| calcofi | 3.83 | 3.98 | 2.85 | **2.77** | 4.53 | 4.75 |
| taxi | 8.67 | 8.25 | **5.22** | 5.27 | 12.4 | 10.1 |

### A.7. Case of ellipsoidal prediction sets

In this section, we will investigate how all parts of our proposed RCP method contribute to the performance. Additionally, we demonstrate the wider applicability of our approach: in this section, we use a different score and ellipsoid prediction sets. To achieve this, we modify our base models to predict parameters of the multivariate normal distribution. As a baseline method, we have selected CQR because of its popularity and ease of use.

Figure 11 demonstrates improvements of `RCP` over simpler methods, with CQR serving as a strong contemporary alternative. Each of these simpler methods employs a consecutively more complex model and/or calibration procedure. The first part of the name corresponds to the base prediction model, and the second part after the dash denotes the calibration procedure:

- Const: the base model for these methods is a constant prediction of multivariate normal distribution parameters estimated on the train set.

- MLP: the base model is a multidimensional perceptron that predicts the parameters of multivariate normal distribution.

- CP: classic conformal prediction using full calibration set. We estimate a fixed prediction ellipsoid size using Mahalanobis distance-based nonconformity score.

- RCP: our usual RCP procedure where we split the calibration set and fit a quantile regression model to predict the $(1 - \alpha)$ quantile of the Mahalanobis score. Similarly to the other experiments, quantile regression is fit using MLP underlying model.

The alternative method CQR is based on quantile regression estimates for each dimension of the output variable. First, (univariate) conformalized quantile regression scores (Romano et al., 2019) are computed for each dimension. Then, they are aggregated by taking the maximum score over each dimension, similarly to ResCP in the main text. The resulting prediction set in this case is a hyperrectangle. Its size is adaptive to the input, but the conformal correction is isotropic and constant for all input points.

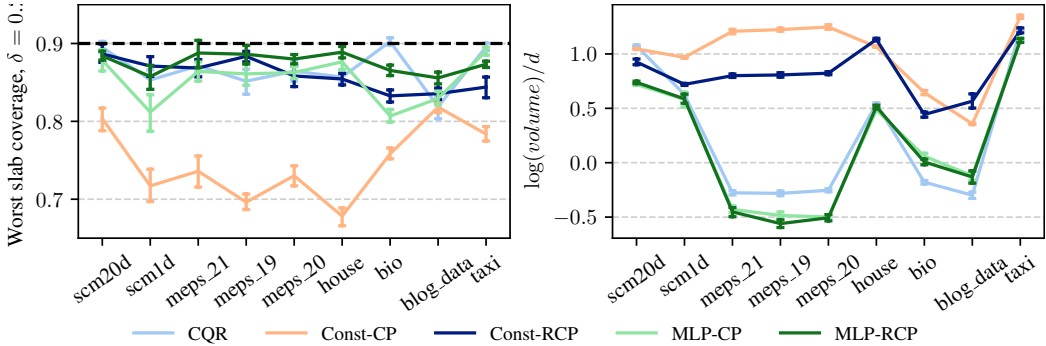

Figure 11: Worst slab coverage and logarithm of prediction set volume (divided by number of dimensions of the response).

The graphs on Figure 11 provide some important insights:

- Methods based on classic conformal prediction (Const-CP, MLP-CP) often struggle to maintain conditional coverage.

- `RCP` improves conditional coverage: Const-RCP outperforms Const-CP in conditional coverage and set size.

- `RCP` in combination with a better predictive model either maintains or improves conditional coverage and volume.

## A.8. Improved data efficiency using cross-validation

As explained in Section 7, `RCP` requires to divide the calibration dataset $\mathcal{D}$ into two parts, one to estimate $\widehat{\tau}$, and one for SCP.

In this section, we consider a more data-efficient approach using the training dataset $\mathcal{D}_{\text{train}}$. Using $K$-fold cross-validation on $\mathcal{D}_{\text{train}}$, for each fold index $k$, we train a model on the $K-1$ remaining folds and evaluate the conformity score on the fold $k$. This yields a dataset $\mathcal{D}_{\tau}$ of size $|\mathcal{D}_{\text{train}}|$ with inputs and their associated conformity scores based on which $\widehat{\tau}$ is estimated. This also removes the need to split the calibration dataset. An additional model is fitted on the complete training data set $\mathcal{D}_{\text{train}}$ to produce the non-rectified conformity scores.

Figure 12 shows a comparison of learning $\widehat{\tau}$ on half the calibration dataset (cal), or using 10-fold cross-validation (CV). The cross-validation approach yields improved worst-slab coverage on most datasets. This improved conditional coverage comes at the computational cost of training $K$ additional models.

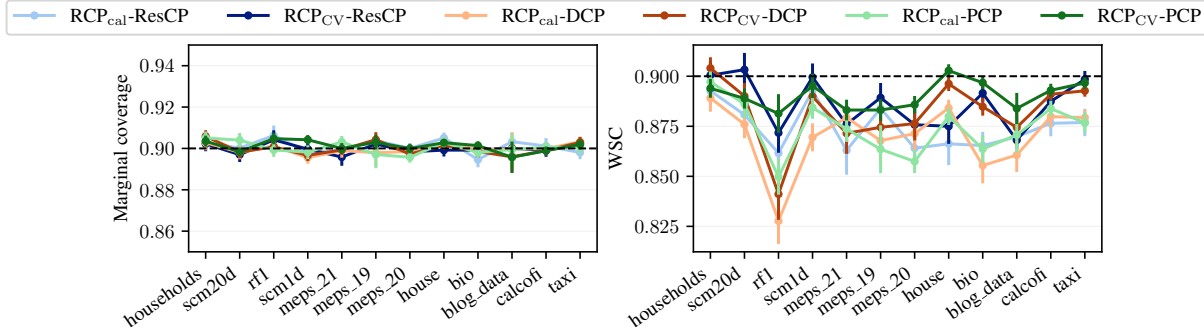

Figure 12: Worst-slab coverage of `RCP` with $\widehat{\tau}$ trained on half the calibration dataset (cal) or using 10-fold cross-validation (CV).

## A.9. Comparison with CPCG

We conduct an additional experiment comparing RCP with Conditional Prediction with Conditional Guarantees (CPCG; Gibbs et al. (2025)). We evaluate RCP using both the full calibration dataset ($RCP_{cal}$) and cross-validation ($RCP_{CV}$), as described in Appendix A.8. All methods are run on CPU (AMD Ryzen Threadripper PRO 5965WX) with 6 CPU threads per experiment.

Table 4 shows that all methods achieve comparable worst-slab coverage, close to the nominal level. However, Table 5 reveals a stark contrast in computational efficiency: CPCG is 200-100,000 times slower than $RCP_{cal}$ and 10-100 times slower than $RCP_{CV}$ overall. This significant overhead is because CPCG must solve an optimization problem involving the entire calibration set *for each test instance*. Consequently, CPCG's computational demands become prohibitive for large calibration and test sets, hindering its practical application. Moreover, CPCG failed to find a solution on the "house" and "calcofi" datasets, precluding results for these cases. These factors highlight RCP's substantial practical advantage in efficiency, especially for large-scale datasets.

Table 4: Comparison of worst-slab coverage on multi-output datasets.

| | PCP | $RCP_{cal}$-PCP | $RCP_{CV}$-PCP | CPCG-PCP | DCP | $RCP_{cal}$-DCP | $RCP_{CV}$-DCP | CPCG-DCP |
|---|---|---|---|---|---|---|---|---|
| households | 0.825 | 0.905 | 0.899 | 0.888 | 0.853 | 0.891 | 0.900 | 0.900 |
| scm20d | 0.830 | 0.892 | 0.891 | 0.897 | 0.877 | 0.877 | 0.868 | 0.899 |
| rf1 | 0.731 | 0.830 | 0.877 | 0.838 | 0.715 | 0.863 | 0.827 | 0.872 |
| scm1d | 0.758 | 0.882 | 0.895 | 0.910 | 0.756 | 0.902 | 0.896 | 0.882 |
| meps_21 | 0.739 | 0.874 | 0.904 | 0.881 | 0.789 | 0.881 | 0.879 | 0.905 |
| meps_19 | 0.762 | 0.875 | 0.867 | 0.880 | 0.788 | 0.884 | 0.889 | 0.878 |
| meps_20 | 0.731 | 0.842 | 0.871 | 0.890 | 0.719 | 0.880 | 0.884 | 0.892 |
| house | 0.835 | 0.895 | 0.903 | / | 0.817 | 0.878 | 0.906 | / |
| bio | 0.784 | 0.860 | 0.900 | 0.887 | 0.774 | 0.879 | 0.880 | 0.880 |
| blog_data | 0.770 | 0.877 | 0.893 | 0.886 | 0.749 | 0.844 | 0.888 | 0.888 |
| calcofi | 0.810 | 0.889 | 0.888 | / | 0.828 | 0.885 | 0.892 | / |
| taxi | 0.837 | 0.885 | 0.884 | 0.881 | 0.846 | 0.872 | 0.879 | 0.879 |

Table 5: Comparison of computational time (in seconds) on multi-output datasets.

| | PCP | $RCP_{cal}$-PCP | $RCP_{CV}$-PCP | CPCG-PCP | DCP | $RCP_{cal}$-DCP | $RCP_{CV}$-DCP | CPCG-DCP |
|---|---|---|---|---|---|---|---|---|
| households | 0.258 | 0.604 | 104 | 8840 | 0.00759 | 0.531 | 104 | 8164 |
| scm20d | 2.63 | 4.22 | 772 | 6409 | 0.0182 | 0.852 | 766 | 6012 |
| rf1 | 0.667 | 1.35 | 340 | 10682 | 0.00836 | 0.348 | 339 | 9674 |
| scm1d | 2.27 | 3.57 | 1209 | 4971 | 0.0133 | 0.867 | 1205 | 4692 |
| meps_21 | 0.236 | 0.607 | 581 | 6283 | 0.0123 | 0.261 | 581 | 6031 |
| meps_19 | 0.272 | 0.515 | 493 | 6411 | 0.0123 | 0.184 | 492 | 6128 |
| meps_20 | 0.255 | 0.520 | 621 | 7147 | 0.0119 | 0.238 | 621 | 7032 |
| house | 0.315 | 0.594 | 1034 | / | 0.0159 | 0.327 | 1033 | / |
| bio | 0.630 | 1.22 | 3161 | 79422 | 0.0279 | 0.782 | 3163 | 63178 |
| blog_data | 0.752 | 0.850 | 1119 | 41192 | 0.0336 | 0.155 | 1121 | 43289 |
| calcofi | 0.699 | 1.02 | 456 | / | 0.0356 | 0.200 | 455 | / |
| taxi | 0.680 | 1.17 | 866 | 77828 | 0.0269 | 0.276 | 866 | 70139 |

# B. Proofs

## B.1. Proof for the first example

We provide here a completely elementary proof. The result actually follows from Theorem 4. In this example, we set $\tau_\star(x) = Q_{1-\alpha}(P_{V|X=x})$, where $\mathbf{V} = V(X, Y)$. We assume that for all $x \in \mathcal{X}$, $\tau_\star(x) > 0$. We denote $\tilde{V}(x, y) = V(x, y)/\tau_\star(x)$ and $\tilde{\mathbf{V}} = \tilde{V}(X, Y)$.

$$Q_{1-\alpha}(P_{\tilde{\mathbf{V}}}) = \inf\{t \in \mathbb{R} : \mathbb{P}(V(X, Y) \le t\tau_\star(X)) \ge 1 - \alpha\}.$$

We will first prove that, for all $x \in \mathcal{X}$, we get that $1 = Q_{1-\alpha}(P_{\mathbf{V}|X=x})$, for all $x \in \mathcal{X}$:

$$\begin{aligned}
Q_{1-\alpha}(P_{\mathbf{V}|X=x}) &= \inf\{t \in \mathbb{R} : \mathbb{P}(V(X, Y) \le t\tau_\star(X)|X = x) \ge 1 - \alpha\} \\
&= \inf\{t \in \mathbb{R} : P_{\mathbf{V}|X=x}((-\infty, tQ_{1-\alpha}(P_{\mathbf{V}|X=x})]) \ge 1 - \alpha\} = 1.
\end{aligned}$$

We then show that $Q_{1-\alpha}(P_{\tilde{\mathbf{V}}}) \le 1$. Indeed, for any $s > 1$, by the tower property of conditional expectation, we get:

$$\begin{aligned}
\mathbb{P}(V(X, Y) \le s\tau_\star(X)) &= \mathbb{P}(\mathbf{V} \le sQ_{1-\alpha}(P_{\mathbf{V}|X})) \\
&= \mathbb{E}[\mathbb{P}(\mathbf{V} \le sQ_{1-\alpha}(P_{\mathbf{V}|X})|X)] \ge 1 - \alpha.
\end{aligned}$$

Assume now that $Q_{1-\alpha}(P_{\tilde{\mathbf{V}}}) < 1$. Then for any $s \in (Q_{1-\alpha}(P_{\tilde{\mathbf{V}}}), 1)$, using again the tower property of conditional expectation, we get

$$\begin{aligned}
1 - \alpha &\le \mathbb{P}(V(X, Y) \le s\tau_\star(X)) = \mathbb{E}[\mathbb{P}(\mathbf{V} \le sQ_{1-\alpha}(P_{\mathbf{V}|X})|X)] && (18) \\
&= \mathbb{E}[P_{\mathbf{V}|X}((-\infty, sQ_{1-\alpha}(P_{\mathbf{V}|X})))] < 1 - \alpha && (19)
\end{aligned}$$

by the definition of the conditional quantile. This yields a contradiction. Therefore, for $P_X$-a.e. $x \in \mathcal{X}$,

$$Q_{1-\alpha}(P_{\tilde{\mathbf{V}}}) = Q_{1-\alpha}(P_{\tilde{\mathbf{V}}|X=x}).$$

## B.2. Proof for the second example

We set in this case $\tilde{V}(x, y) = V(x, y) - \tau_\star(x)$, where $\tau_\star(x) = Q_{1-\alpha}(P_{\mathbf{V}|X=x})$ and $\tilde{\mathbf{V}} = \tilde{V}(X, Y)$. We will show that $Q_{1-\alpha}(P_{\tilde{\mathbf{V}}|X=x}) = 0$ for all $x \in \mathcal{X}$. We have indeed:

$$\begin{aligned}
Q_{1-\alpha}(P_{\tilde{\mathbf{V}}|X=x}) &= \inf\{t \in \mathbb{R} : \mathbb{P}(\tilde{V}(X, Y) \le t|X = x) \ge 1 - \alpha\} && (20) \\
&= \inf\{t \in \mathbb{R} : \mathbb{P}(V(X, Y) \le \tau_\star(X) + t|X = x) \ge 1 - \alpha\} = 0. && (21)
\end{aligned}$$

We will now show that $Q_{1-\alpha}(P_{\tilde{\mathbf{V}}}) \le 0$. Indeed, for all $s > 0$, by the tower property of conditional expectation and the definition of the conditional quantile, we get

$$\mathbb{P}(\tilde{V}(X, Y) \le s) = \mathbb{E}[\mathbb{P}(V(X, Y) \le \tau_\star(X) + s|X)] \ge 1 - \alpha. \tag{22}$$

On the other hand, assume $Q_{1-\alpha}(P_{\tilde{\mathbf{V}}}) < 0$. Set $s \in (Q_{1-\alpha}(\mathbb{P}_{\tilde{\mathbf{V}}}, 0)$. We get

$$\begin{aligned}
1 - \alpha &\le \mathbb{P}(\tilde{V}(X, Y) \le s) = \mathbb{P}(V(X, Y) \le s + \tau_\star(X)) && (23) \\
&= \mathbb{E}[\mathbb{P}(V(X, Y) \le s + \tau(X)|X)] < 1 - \alpha, && (24)
\end{aligned}$$

which leads to a contradiction.

We first show that $Q_{1-\alpha}(\mathbb{P}_{\tilde{V}}) \le 0$. Indeed, by the tower property of conditional expectation, using again the definition of the conditional quantile, we get

$$\mathbb{P}(\tilde{V}(X, Y) \le 0) = \mathbb{E}[\mathbb{P}(V(X, Y) \le \tau(X)|X)] < 1 - \alpha, \tag{25}$$

which leads to a contradiction and concludes the proof.

**B.3. Proof of Theorem 1**

We will now proceed with the proof of Theorem 1, which verifies the marginal validity of our proposed approach. First, recall that $V \colon \mathcal{X} \times \mathcal{Y} \to \mathbb{R}$ is a conformity score function to which we apply a measurable transformation $(t, x) \in \mathbb{T} \times \mathcal{X} \to f_t^{-1}(x)$. Recall that `RCP` constructs the following prediction sets for $x \in \mathcal{X}$

$$\mathcal{C}_\alpha(x) = \left\{ y \in \mathcal{Y} \colon f_{\tau(x)}^{-1} \circ V(x, y) \leq Q_{(1-\alpha)(1+n^{-1})} \left( \frac{1}{n} \sum_{k=1}^n \delta_{f_{\tau(X_k)}^{-1} \circ V(X_k, Y_k)} \right) \right\}.$$

For any $k \in \{1, \dots, n+1\}$, denote $\tilde{V}_k = f_{\tau(X_k)}^{-1} \circ V(X_k, Y_k)$.

**Theorem 3.** *Assume **H1-H2** hold, and let $\alpha \in [\{n+1\}^{-1}, 1)$. If $\tilde{V}_1, \dots, \tilde{V}_{n+1}$ are almost surely distinct, then it yields*

$$1 - \alpha \leq \mathbb{P}\left(Y_{n+1} \in \mathcal{C}_\alpha(X_{n+1})\right) < 1 - \alpha + \frac{1}{n+1}. \tag{26}$$

*Proof.* By definition, we have

$$\mathbb{P}\left(Y_{n+1} \in \mathcal{C}_\alpha(X_{n+1})\right) = \mathbb{P}\left(f_{\tau(X_{n+1})}^{-1} \circ V(X_{n+1}, Y_{n+1}) \leq Q_{(1-\alpha)(1+n^{-1})}(\tfrac{1}{n}\textstyle\sum_{k=1}^n \delta_{\tilde{V}_k})\right) \tag{27}$$

$$= \mathbb{P}\left(\tilde{V}_{n+1} \leq Q_{(1-\alpha)(1+n^{-1})}(\tfrac{1}{n}\textstyle\sum_{k=1}^n \delta_{\tilde{V}_k})\right). \tag{28}$$

Denote by $F_{\tilde{V}}$ the cumulative density function of $f_{\tau(X_{n+1})}^{-1} \circ V(X_{n+1}, Y_{n+1})$ and consider $\{U_1, \dots, U_{n+1}\}$ a family of mutually independent uniform random variables. Given $\alpha \in [\{n+1\}^{-1}, 1)$, define

$$k_\alpha = \lceil n(1 + n^{-1})(1 - \alpha) \rceil.$$

Since by assumption $\alpha \geq \{n+1\}^{-1}$, we have $k_\alpha \in \{1, \dots, n\}$. Additionally, remark that $\tilde{V}_k$ has the same distribution that $F_{\tilde{V}}^{-1}(U_k)$. Therefore, by independence of the data, we can write

$$\mathbb{P}\left(\tilde{V}_{n+1} \leq Q_{(1-\alpha)(1+n^{-1})}(\tfrac{1}{n}\textstyle\sum_{k=1}^n \delta_{\tilde{V}_k})\right) = \mathbb{P}\left(F_{\tilde{V}}^{-1}(U_{n+1}) \leq F_{\tilde{V}}^{-1}(U_{(k_\alpha)})\right),$$

where $U_{(1)}, \dots, U_{(n)}$ denotes the order statistics. Additionally, since the scores $\tilde{V}_1, \dots, \tilde{V}_{n+1}$ are almost surely distinct, we deduce that

$$\mathbb{P}\left(F_{\tilde{V}}^{-1}(U_{n+1}) \leq F_{\tilde{V}}^{-1}(U_{(k_\alpha)})\right) = \mathbb{P}\left(U_{n+1} \leq U_{(k_\alpha)}\right) = \mathbb{E}\left[U_{(k_\alpha)}\right].$$

Since $U_{(k_\alpha)}$ follows a beta distribution with parameters $(k_\alpha, n+1-k_\alpha)$, we obtain that $\mathbb{E}\left[U_{(k_\alpha)}\right] = (n+1)^{-1}k_\alpha$. $\qquad\square$

**B.4. Proof of equality** (9)

**Theorem 4.** *Assume **H1-H2** hold. For $x \in \mathcal{X}$, set $\tau_\star(x) = Q_{1-\alpha}(\mathrm{P}_{\mathbf{V}_\varphi \mid X=x})$, where $V_\varphi(x, y) = \tilde{f}_\varphi^{-1} \circ V(x, y)$. Set $\tilde{V}_\varphi(x, y) = f_{\tau_\star(x)}^{-1} \circ V(x, y)$ and $\tilde{\mathbf{V}}_\varphi = \tilde{V}_\varphi(X, Y)$. Then, for all $x \in \mathcal{X}$,*

$$\varphi = Q_{1-\alpha}(\mathrm{P}_{\tilde{\mathbf{V}}_\varphi \mid X=x}) = Q_{1-\alpha}(\mathrm{P}_{\tilde{\mathbf{V}}_\varphi}).$$

*Proof.* Set $\psi(x) = Q_{1-\alpha}(\mathrm{P}_{\tilde{\mathbf{V}}_\varphi \mid X=x})$. We must prove that $\psi(x) = \phi$ for all $x \in \mathcal{X}$. First, we will show $\psi(x) \leq \varphi$. Note indeed

$$\mathbb{P}(\tilde{V}_\varphi(X, Y) \leq \varphi \mid X = x) = \mathbb{P}(V(X, Y) \leq f_{\tau_\star(X)}(\varphi) \mid X = x) \overset{(a)}{=} \mathbb{P}(V(X, Y) \leq \tilde{f}_\varphi(\tau_\star(X)) \mid X = x) \tag{29}$$

$$\overset{(b)}{=} \mathbb{P}(\tilde{f}_\varphi^{-1} \circ V(X, Y) \leq \tau_\star(X) \mid X = x) \overset{(c)}{\geq} 1 - \alpha, \tag{30}$$

where (a) follows from $f_t(\varphi) = \tilde{f}_\varphi(t)$, (b) from the fact that $\tilde{f}_\varphi$ is invertible, and (c) from the definition of $\tau_\star(x)$.

Now, suppose that $\psi(x) < \varphi$. Since for any $t$, $f_t$ is increasing, we get that $f_{\tau(x)}(\psi(x)) < f_{\tau(x)}(\varphi)$. Moreover, using that $\tau(x)$ belongs to the interior of $\mathbb{T}$, combined with the continuity of $t \in \mathbb{T} \mapsto \tilde{f}_\varphi(t)$; it implies the existence of $\tilde{t} \in \mathbb{T}$ such that $\tilde{t} < \tau(x)$ and also $f_{\tau(x)}(\psi(x)) < f_{\tilde{t}}(\varphi)$. We can rewrite

$$1 - \alpha \leq \mathbb{P}\left(V(X,Y) \leq f_{\tau_\star(X)}(\psi(X)) \mid X = x\right) \leq \mathbb{P}\left(V(X,Y) \leq f_{\tilde{t}}(\varphi) \mid X = x\right)$$
$$= \mathbb{P}\left(\tilde{f}_\varphi^{-1} \circ V(X,Y) \leq \tilde{t} \mid X = x\right) < 1 - \alpha.$$

which yields to a contradiction.

We now show that $Q_{1-\alpha}(P_{\tilde{\mathbf{V}}_\varphi}) = \varphi$. We first show that $Q_{1-\alpha}(P_{\tilde{\mathbf{V}}_\varphi}) = \varphi$. We first show that $Q_{1-\alpha}(P_{\tilde{\mathbf{V}}_\varphi}) \leq \varphi$. This follows from

$$\mathbb{P}(\tilde{V}_\varphi(X,Y) \leq \varphi) \overset{(a)}{=} \mathbb{E}[\mathbb{P}(\tilde{V}_\varphi(X,Y) \leq \varphi|X)] \overset{(b)}{\geq} 1 - \alpha,$$

where (a) follows from the tower property of conditional expectation and (b) from $\phi = Q_{1-\alpha}(P_{\tilde{\mathbf{V}}_\varphi|X=x})$ for all $x \in \mathcal{X}$.

Assume now that $Q_{1-\alpha}(P_{\tilde{\mathbf{V}}_\varphi}) < \varphi$. Choose $s \in (Q_{1-\alpha}(P_{\tilde{\mathbf{V}}_\varphi}), \varphi)$. Then,

$$1 - \alpha \leq \mathbb{P}(\tilde{V}_\varphi(X,Y) \leq s) \overset{(a)}{=} \mathbb{E}[\mathbb{P}(\tilde{V}_\varphi(X,Y) \leq s|X)] \overset{(b)}{<} 1 - \alpha,$$

where (a) follows from the tower property of conditional expectation and (b) $s < \phi = Q_{1-\alpha}(P_{\tilde{\mathbf{V}}_\varphi|X=x})$ for all $x \in \mathcal{X}$. This yields to a contradiction which concludes the proof. $\qquad\square$

### B.5. Proof of Theorem 2

This section is devoted to the proof of the conditional guarantee given in Section 6. In this section, we denote $\tilde{V}(x,y) = f_{\hat{\tau}(x)}^{-1}(V(x,y))$ and for each $t \in \mathbb{R}$, we denote

$$F_{\tilde{\mathbf{V}}|X=x}(t) = \mathbb{P}(\tilde{V}(X,Y) \leq t \mid X = x) \qquad \text{and} \qquad F_{\tilde{\mathbf{V}}}(t) = \mathbb{P}(\tilde{V}(X,Y) \leq t).$$

For any $x \in \mathcal{X}$, we assess the quality of the quantile estimate $\tau(x)$ via

$$\epsilon_\tau(x) = \mathbb{P}\left(\tilde{f}_\varphi^{-1}(V(x,Y)) \leq \hat{\tau}(x) \mid X = x\right) - 1 + \alpha.$$

For all $n \in \mathbb{N}$, note that $\alpha(1-\alpha)^{n+1} \leq \frac{e}{n+2}$. If $\alpha \geq 0.1$ and $n \geq 100$, then $\alpha(1-\alpha)^{n+1} \leq \frac{1}{4183n}$. In addition, if $F_{\tilde{V}}(\varphi) \leq 1 - \alpha$, then $\alpha L F_{\tilde{V}}(\varphi) \leq \frac{L}{4183n}$.

**Theorem 5.** *Assume that **H1-H2** hold. Assume in addition that, for any $x \in \mathcal{X}$, $F_{\tilde{V}}$ is continuous and $F_{\tilde{V}|X=x} \circ F_{\tilde{V}}^{-1}$ is* L-*Lipschitz. Then for $\alpha \in [\{n+1\}^{-1}, 1)$ it holds*

$$1 - \alpha + \epsilon_\tau(x) - \alpha L \times [F_{\tilde{V}}(\varphi)]^{n+1} \leq \mathbb{P}\left(Y_{n+1} \in \mathcal{C}_\alpha(X_{n+1}) \mid X_{n+1} = x\right)$$
$$\leq 1 - \alpha + \epsilon_\tau(x) + L(1 - \alpha + (n+1)^{-1}) \times [1 - F_{\tilde{V}}(\varphi)]^{n+1}.$$

*Proof.* Let $t \in \mathbb{R}$ be fixed. A first calculation shows that

$$\mathbb{P}\left(f_{\tau(X)}^{-1}(V(X,Y)) \leq t\right) = \int \mathbb{P}\left(f_{\tau(x)}^{-1}(V(x,Y)) \leq t \mid X = x\right) P_X(\mathrm{d}x)$$
$$= \int \mathbb{P}\left(\tilde{f}_t^{-1}(V(x,Y)) \leq \tau(x) \mid X = x\right) P_X(\mathrm{d}x).$$

Now, we introduce the notation $\Delta_t(x)$, which quantifies the discrepancy between substituting $\varphi$ with $t$:

$$\Delta_t(x) = \mathbb{P}\left(\tilde{f}_t^{-1}(V(x,Y)) \leq \tau(x) \mid X = x\right) - \mathbb{P}\left(\tilde{f}_\varphi^{-1}(V(x,Y)) \leq \tau(x) \mid X = x\right).$$

Let's $P_Q$ denote the distribution of the empirical quantile $Q_{(1-\alpha)(1+n^{-1})}(\frac{1}{n}\sum_{k=1}^{n}\delta_{\tilde{V}_k})$. We can rewrite the conditional coverage as follows

$$\mathbb{P}\left(Y \in \mathcal{C}_\alpha(X) \mid X = x\right) = \int \mathbb{P}\left(f_{\tau(x)}^{-1}(V(x,Y)) \leq t \mid X = x\right) \mathrm{P}_Q(\mathrm{d}t)$$

$$= \int \mathbb{P}\left(\tilde{f}_t^{-1}(V(x,Y)) \leq \tau(x) \mid X = x\right) \mathrm{P}_Q(\mathrm{d}t)$$

$$= \mathbb{P}\left(\tilde{f}_\varphi^{-1}(V(x,Y)) \leq \tau(x) \mid X = x\right) + \int \Delta_t(x)\mathrm{P}_Q(\mathrm{d}t)$$

$$= 1 - \alpha + \epsilon_\tau(x) + \int \Delta_t(x)\mathrm{P}_Q(\mathrm{d}t).$$

Moreover, consider a set of $n$ i.i.d. uniform random variables $\{U_k\}_{1 \leq k \leq n}$, and let $U_{(1)} \leq \ldots \leq U_{(n)}$ denote their order statistics. Since $\tilde{V}_1, \ldots, \tilde{V}_n$ are i.i.d., their joint distribution is the same as $(F_{\tilde{V}}^{-1}(U_1), \ldots, F_{\tilde{V}}^{-1}(U_n))$. Therefore, $\mathrm{P}_Q$ is also the distribution of the $(1+n^{-1})(1-\alpha)$-quantile of $\frac{1}{n}\sum_{k=1}^{n}\delta_{F_{\tilde{V}}^{-1}(U_k)}$. Thus, there exists an integer $k_\alpha \in \{1, \ldots, n\}$ such that

$$F_{\tilde{V}}^{-1}(U_{(k_\alpha)}) = Q_{(1-\alpha)(1+n^{-1})}\left(\frac{1}{n}\sum_{k=1}^{n}\delta_{F_{\tilde{V}}^{-1}(U_k)}\right).$$

Moreover, using that $\{\tilde{V}_k \colon k \in [n]\}$ are almost surely distinct, we deduce the existence of the minimal integer $k_\alpha \in [n]$ such that

$$\frac{1}{n}\sum_{k=1}^{n}\mathbb{1}_{U_{(k)} \leq U_{(k_\alpha)}} \geq \left(1 + \frac{1}{n}\right)(1-\alpha).$$

Since $\sum_{k=1}^{n}\mathbb{1}_{U_{(k)} \leq U_{(k_\alpha)}} = k_\alpha$ almost surely, we deduce that $k_\alpha = \lceil (n+1)(1-\alpha) \rceil$. We also get that $F_{\tilde{V}}^{-1}(U_{(k_\alpha)}) \sim \mathrm{P}_Q$. In the following, we provide a lower bound on $\Delta_t(x)$. Since $F_{\tilde{V}|X=x}$ is increasing, we can write

$$\int \Delta_t(x)\mathrm{P}_Q(\mathrm{d}t) = \mathbb{E}\left[F_{\tilde{V}|X=x} \circ F_{\tilde{V}}^{-1}(U_{(k_\alpha)}) - F_{\tilde{V}|X=x}(\varphi)\right]. \tag{31}$$

**Lower bound.** First, using (31) implies that

$$\int \Delta_t(x)\mathrm{P}_Q(\mathrm{d}t) \geq -\mathbb{E}\left[\mathbb{1}_{\varphi \geq F_{\tilde{V}}^{-1}(U_{(k_\alpha)})} \times \left\{F_{\tilde{V}|X=x}(\varphi) - F_{\tilde{V}|X=x} \circ F_{\tilde{V}}^{-1}(U_{(k_\alpha)})\right\}_+\right].$$

Moreover, by definition of the cumulative density function and its inverse, we have $F_{\tilde{V}|X=x} \circ F_{\tilde{V}}^{-1} \circ F_{\tilde{V}}(\varphi) \leq F_{\tilde{V}|X=x}(\varphi)$. Thus, it follows that

$$\mathbb{E}\left[\mathbb{1}_{\varphi \geq F_{\tilde{V}}^{-1}(U_{(k_\alpha)})} \times \left\{F_{\tilde{V}|X=x}(\varphi) - F_{\tilde{V}|X=x} \circ F_{\tilde{V}}^{-1}(U_{(k_\alpha)})\right\}_+\right]$$

$$\leq F_{\tilde{V}|X=x}(\varphi) - F_{\tilde{V}|X=x} \circ F_{\tilde{V}}^{-1} \circ F_{\tilde{V}}(\varphi)$$

$$+ \mathbb{E}\left[\mathbb{1}_{\varphi \geq F_{\tilde{V}}^{-1}(U_{(k_\alpha)})} \times \left\{F_{\tilde{V}|X=x} \circ F_{\tilde{V}}^{-1} \circ F_{\tilde{V}}(\varphi) - F_{\tilde{V}|X=x} \circ F_{\tilde{V}}^{-1}(U_{(k_\alpha)})\right\}_+\right]. \tag{32}$$

If $F_{\tilde{V}}(\varphi) = 1$, then $F_{\tilde{V}|X=x}(\varphi) = 1$ $\mathrm{P}_X$-almost everywhere. Let's now suppose that $F_{\tilde{V}}(\varphi) < 1$ and let's define $\varphi_\star = \sup\{t \in \mathbb{R} \colon F_{\tilde{V}}(t) = F_{\tilde{V}}(\varphi)\}$. For any $\epsilon > 0$, note that $F_{\tilde{V}}(\varphi_\star + \epsilon) > F_{\tilde{V}}(\varphi_\star)$. This leads to

$$F_{\tilde{V}}^{-1} \circ F_{\tilde{V}}(\varphi_\star + \epsilon) = \inf\{t \in \mathbb{R} \colon F_{\tilde{V}}(t) \geq F_{\tilde{V}}(\varphi_\star + \epsilon)\} > \varphi_\star.$$

Furthermore, using the L-Lipschitz assumption on $F_{\tilde{V}|X=x} \circ F_{\tilde{V}}^{-1}$ implies that

$$0 \leq F_{\tilde{V}|X=x}(\varphi) - F_{\tilde{V}|X=x} \circ F_{\tilde{V}}^{-1} \circ F_{\tilde{V}}(\varphi)$$

$$\leq \liminf_{\epsilon \to 0_+}\left\{F_{\tilde{V}|X=x} \circ F_{\tilde{V}}^{-1} \circ F_{\tilde{V}}(\varphi_\star + \epsilon) - F_{\tilde{V}|X=x} \circ F_{\tilde{V}}^{-1} \circ F_{\tilde{V}}(\varphi)\right\}$$

$$\leq \mathrm{L}\liminf_{\epsilon \to 0_+}\left\{F_{\tilde{V}}(\varphi_\star + \epsilon) - F_{\tilde{V}}(\varphi)\right\}. \tag{33}$$

From the continuity of $F$, we deduce that $F_{\tilde{V}}(\varphi) = F_{\tilde{V}}(\varphi_\star)$. Therefore, we can conclude that $\liminf_{\epsilon \to 0_+} \{F_{\tilde{V}}(\varphi_\star + \epsilon) - F_{\tilde{V}}(\varphi)\} = 0$. This computation combined with (33) shows that $F_{\tilde{V}|X=x}(\varphi) = F_{\tilde{V}|X=x} \circ F_{\tilde{V}}^{-1} \circ F_{\tilde{V}}(\varphi)$. Lastly, it just remains to upper bound the last term of (32). Once again, using that $F_{\tilde{V}|X=x} \circ F_{\tilde{V}}^{-1}$ is Lipschitz gives

$$\mathbb{E}\left[\mathbb{1}_{\varphi \geq F_{\tilde{V}}^{-1}(U_{(k_\alpha)})} \times \left\{F_{\tilde{V}|X=x} \circ F_{\tilde{V}}^{-1} \circ F_{\tilde{V}}(\varphi) - F_{\tilde{V}|X=x} \circ F_{\tilde{V}}^{-1}(U_{(k_\alpha)})\right\}_+\right] \leq L\mathbb{E}\left[\left\{F_{\tilde{V}}(\varphi) - U_{(k_\alpha)}\right\}_+\right].$$

Finally, applying Lemma 1 with $\beta = F_{\tilde{V}}(\varphi)$ and $k = k_\alpha$ yields the lower bound.

**Upper bound.** From (31), we deduce that

$$\int \Delta_t(x) \mathrm{P}_Q(\mathrm{d}t) \leq \mathbb{E}\left[\mathbb{1}_{\varphi \leq F_{\tilde{V}}^{-1}(U_{(k_\alpha)})} \times \left\{F_{\tilde{V}|X=x} \circ F_{\tilde{V}}^{-1}(U_{(k_\alpha)}) - F_{\tilde{V}|X=x}(\varphi)\right\}_+\right]. \tag{34}$$

By definition of $F_{\tilde{V}}^{-1}$, we get $\varphi \geq F_{\tilde{V}}^{-1} \circ F_{\tilde{V}}(\varphi)$. Since $F_{\tilde{V}|X=x}$ is increasing and $F_{\tilde{V}|X=x} \circ F_{\tilde{V}}^{-1}$ is L-Lipschitz, it follows that

$$\begin{aligned}
\int \Delta_t(x) \mathrm{P}_Q(\mathrm{d}t) &\leq \mathbb{E}\left[\mathbb{1}_{\varphi \leq F_{\tilde{V}}^{-1}(U_{(k_\alpha)})} \times \left\{F_{\tilde{V}|X=x} \circ F_{\tilde{V}}^{-1}(U_{(k_\alpha)}) - F_{\tilde{V}|X=x}(\varphi)\right\}_+\right] \\
&\leq L\mathbb{E}\left[\left\{U_{(k_\alpha)} - F_{\tilde{V}}(\varphi)\right\}_+\right] \\
&= L\mathbb{E}\left[\left\{1 - F_{\tilde{V}}(\varphi) - (1 - U_{(k_\alpha)})\right\}_+\right].
\end{aligned}$$

Since the distribution of $1 - U_{(k_\alpha)}$ is the same that the distribution of $U_{(n+1-k_\alpha)}$, applying Lemma 1 with $\beta = 1 - F_{\tilde{V}}(\varphi)$ and $k = n + 1 - k_\alpha$ yields the upper bound. $\qquad\square$

Let's denote by $U_{(k)}$ the $k$th order statistic of the i.i.d. uniform random variables $U_1, \ldots, U_n$.

**Lemma 1.** *For any $\beta \in [0, 1]$ and $k \in [n]$, it holds that*

$$\mathbb{E}\left[(\beta - U_{(k)})_+\right] = \beta^{n+1}\left(1 - \frac{k}{n+1}\right).$$

*Proof.* Let $\beta \in [0, 1]$ be fixed. For any $(i, j) \in \mathbb{N}^2$, define

$$I(i, j) = \int_0^\beta u^i(1 - u)^j \mathrm{d}u.$$

By applying integration by parts for $j \geq 1$, we obtain

$$\frac{I(i, j)}{i!j!} = \frac{I(i+1, j-1)}{(i+1)!(j-1)!} = \cdots = \frac{I(i+j, 0)}{(i+j)!} = \frac{\beta^{i+j+1}}{(i+j+1)!}.$$

Since $U_{(k)}$ follows a beta distribution with parameters $(k, n+1-k)$, it follows that

$$\mathbb{E}\left[(\beta - U_{(k)})_+\right] = \int_0^\beta \frac{n!(\beta - u)}{(k-1)!(n-k)!}u^{k-1}(1-u)^{n-k}\mathrm{d}u. \tag{35}$$

Furthermore, we have the following derivations:

$$\begin{aligned}
&\int_0^\beta (\beta - u)u^{k-1}(1-u)^{n-k}\mathrm{d}u \\
&= \beta \int_0^\beta u^{k-1}(1-u)^{n-k}\mathrm{d}u - \int_0^\beta (\beta - u)u^k(1-u)^{n-k}\mathrm{d}u \\
&= \beta I(k-1, n-k) - I(k, n-k). \tag{36}
\end{aligned}$$

Lastly, combining (35) with (36) yields the next result

$$\mathbb{E}\left[(\beta - U_{(k)})_+\right] = \beta \frac{n!I(k-1, n-k)}{(k-1)!(n-k)!} - \frac{n!I(k, n-k)}{(k-1)!(n-k)!} = \beta^{n+1} - \frac{\beta^{n+1}k}{n+1}.$$

$\square$

For any $\beta \in [0, 1]$, observe that

$$(1 - \beta)\beta^{n+1} \leq \frac{\exp\left((n+1)\log(1 - (n+2)^{-1})\right)}{n+2}.$$

Noting that $\log(1 - (n+2)^{-1}) = \sum_{k \geq 1} k^{-1}(n+2)^{-k}$, we can show that:

$$(n+1)\log\left(1 - \frac{1}{n+2}\right) = 1 - \frac{1}{n+2} + \frac{n+1}{(n+2)^2}\sum_{k \geq 0}\frac{(n+2)^{-k}}{k+2} \leq 1.$$

Consequently, this implies that $(1 - \beta)\beta^{n+1} \leq (n+2)^{-1}\mathrm{e}$.

### B.6. Pointwise control of $\epsilon_\tau$

In this section, we control the quality of the $(1 - \alpha)$-conditional quantile estimator $\tau(x)$. To do this, recall that $V_\varphi(x, y) = \tilde{f}_\varphi^{-1} \circ V(x, y)$ and consider the following error

$$\epsilon_\tau(x) = \mathbb{P}\left(V_\varphi(x, Y) \leq \tau(x) \mid X = x\right) - 1 + \alpha.$$

Moreover, in this section we denote by $q_{1-\alpha}(x)$ the conditional $(1 - \alpha)$-quantile of $V_\varphi(x, Y)$ given $X = x$.

**Theorem 6.** *For $x \in \mathcal{X}$, assume that $V_\varphi(x, Y)$ has a 1-st moment. If for any $t \in \mathbb{R}$, $\mathbb{P}(V_\varphi(x, Y) = t \mid X = x) = 0$, then*

$$|\epsilon_\tau(x)| \leq \sqrt{2\left\{\mathcal{L}_x(\tau(x)) - \mathcal{L}_x(q_{1-\alpha}(x))\right\}}.$$

*Proof.* Let $x \in \mathcal{X}$ be fixed. By definition of $\epsilon_\tau$, we can write

$$\mathbb{E}\left[\epsilon_\tau(X)^2\right] = \mathbb{E}\left[\left(\mathbb{P}\left(V_\varphi(x, Y) \leq \tau(x) \mid X = x\right) - 1 + \alpha\right)^2\right].$$

Moreover, for any $t \in \mathbb{R} \setminus \{0\}$, it holds that

$$\rho'_{1-\alpha}(t) = \mathbb{1}_{t \leq 0} - 1 + \alpha.$$

By extension, consider $\rho'_{1-\alpha}(0) = 1$. Hence, we get

$$\begin{aligned}
\epsilon_\tau(x) &= \mathbb{P}\left(V_\varphi(x, Y) \leq \tau(x) \mid X = x\right) - 1 + \alpha \\
&= \mathbb{E}\left[\mathbb{1}_{V_\varphi(x,Y) \leq \tau(x)} \mid X = x\right] - 1 + \alpha \\
&= \mathbb{E}\left[\rho'_{1-\alpha}(V_\varphi(x, Y) - \tau(x)) \mid X = x\right].
\end{aligned}$$

For $t \in \mathbb{R}$, define the loss $\mathcal{L}_x(t)$ as follows

$$\mathcal{L}_x(t) = \mathbb{E}\left[\rho_{1-\alpha}(V_\varphi(x, Y) - t) \mid X = x\right].$$

Since $\mathcal{L}_x(t)$ is convex with Lipshitz continuous gradient, applying Theorem 2.1.5 from (Nesterov, 1998), it follows that

$$\left|\mathcal{L}'_x(t_1) - \mathcal{L}'_x(t_0)\right|^2 \leq 2\mathrm{L} \times D_{\mathcal{L}_x}(t_1, t_0),$$

where L denotes the Lipschitz constant of $\mathcal{L}'_x$, and where $D_{\mathcal{L}_x}$ is the Bregman divergence associated with $\mathcal{L}_x$. For $t_0, t_1 \in \mathbb{R}$, the expression of the Bregman divergence is given by

$$D_{\mathcal{L}_x}(t_1, t_0) = \mathcal{L}_x(t_1) - \mathcal{L}_x(t_0) - \mathcal{L}'_x(t_0)(t_1 - t_0).$$

Let $t_0 = q_{1-\alpha}(x)$, which represents the true quantile. Given that $V_\varphi(x, Y)$ has no probability mass at $q_{1-\alpha}(x)$, we have $\mathcal{L}'_x(q_{1-\alpha}(x)) = 0$. Moreover, by setting $t_1 = \tau(x)$, we can observe that

$$\left|\mathcal{L}'_x(\tau(x))\right|^2 \leq 2L \times \left(\mathcal{L}_x(\tau(x)) - \mathcal{L}_x(q_{1-\alpha}(x))\right).$$

Note that $\mathcal{L}'_x(\tau(x)) = -\epsilon_\tau(x)$, therefore, the previous line shows

$$|\epsilon_\tau(x)| \leq \sqrt{2L \times \{\mathcal{L}_x(\tau(x)) - \mathcal{L}_x(q_{1-\alpha}(x))\}}.$$

Finally, since the derivative of the Pinball loss function is 1-Lipschitz, it follows that $L \leq 1$. $\qquad\square$

In the following, we denote for any $t \in \mathbb{R}$

$$F_{V_\varphi|X=x}(t) = \mathbb{P}\left(\tilde{f}_\varphi^{-1} \circ V(X, Y) \leq t \mid X = x\right).$$

Moreover, let's denote by $\hat{F}_{V_\varphi|X=x}$ an estimator of the cumulative density function $F_{V_\varphi|X=x}$. For $x \in \mathcal{X}$, define

$$\tau(x) = \inf\left\{t \in \mathbb{R}: \hat{F}_{V_\varphi|X=x}(t) \geq 1 - \alpha\right\}.$$

**Lemma 2.** *For $x \in \mathcal{X}$, assume that $\hat{F}_{V_\varphi|X=x}$ is continuous. Then, for any $\alpha \in (0, 1)$,*

$$|\epsilon_\tau(x)| \leq \|F_{V_\varphi|X=x} - \hat{F}_{V_\varphi|X=x}\|_\infty.$$

*Proof.* Let $x$ be in $\mathcal{X}$. Since $\hat{F}_{V_\varphi|X=x}$ is supposed continuous, we have $\hat{F}_{V_\varphi|X=x}(\tau(x)) = 1 - \alpha$. Furthermore, using that $\epsilon_\tau(x) = F_{V_\varphi|X=x}(\tau(x)) - \alpha + 1$, we obtain that

$$
\begin{aligned}
|\epsilon_\tau(x)| &= \left|F_{V_\varphi|X=x} \circ \hat{F}_{V_\varphi|X=x}^{-1}(1 - \alpha) - \alpha + 1\right| \\
&= \left|F_{V_\varphi|X=x} \circ \hat{F}_{V_\varphi|X=x}^{-1}(1 - \alpha) - \hat{F}_{V_\varphi|X=x} \circ \hat{F}_{V_\varphi|X=x}^{-1}(1 - \alpha)\right| \\
&\leq \|F_{V_\varphi|X=x} - \hat{F}_{V_\varphi|X=x}\|_\infty.
\end{aligned}
$$

$\qquad\square$

### B.7. Uniform convergence of cumulative density estimator

For any $k \in [m]$, set $\tilde{V}_{\varphi,k} = \tilde{f}_\varphi^{-1} \circ V(X_k, Y_k)$. In the whole section, we assume that the random variables $X_1, \ldots, X_m$ are i.i.d. Therefore, the random variables $\tilde{w}_k(x) = K_{h_X}(\|x - X_k\|)$ defined for all $k \in [m]$ are mutually independent. Moreover, let's consider the empirical cumulative function given for $x \in \mathcal{X}$ and $v \in \mathbb{R}$, by

$$\hat{F}_{\tilde{V}_\varphi|X}(v \mid x) = \sum_{k=1}^m w_k(x)\mathbb{1}_{\tilde{V}_{\varphi,k} \leq v}.$$

**Theorem 7.** *If H3 holds, then, it holds that*

$$
\mathbb{P}\left(\left\|\hat{F}_{\tilde{V}_\varphi|X}(v \mid x) - F_{\tilde{V}_\varphi|X}(v \mid x)\right\|_\infty \geq \left(\sqrt{\tfrac{2\|K_1\|_\infty}{h_X}} + \sup_{t \in \mathbb{R}_+}\{MtK_1(t)\}\right)\sqrt{\frac{2\log m}{m\mathbb{E}[\tilde{w}_k(x)]^2}} + \frac{2D_{h_X}(x)}{\mathbb{E}\tilde{w}_k(x)}\right)
$$
$$
\leq \frac{2 + 4\mathbb{E}[\tilde{w}_k(x)]^{-1}\operatorname{Var}[\tilde{w}_k(x)]}{m},
$$

*where $D_{h_X}(x)$ is defined in* (46).

*Proof.* Let $x \in \mathcal{X}$ and $v \in \mathbb{R}$ be fixed. First, recall that $F_{\tilde{V}_\varphi | X}(v \mid x) = \mathbb{P}(V(X, Y) \leq v \mid X = x)$. We will now control $\hat{F}_{\tilde{V}_\varphi | X}(v \mid x) - F_{\tilde{V}_\varphi | X}(v \mid x)$ as below:

$$\hat{F}_{\tilde{V}_\varphi | X}(v \mid x) - F_{\tilde{V}_\varphi | X}(v \mid x) = \sum_{k=1}^{m} w_k(x) \left\{ \mathbb{1}_{\tilde{V}_{\varphi,k} \leq v} - F_{\tilde{V}_\varphi | X}(v \mid X_k) \right\}$$

$$+ \sum_{k=1}^{m} w_k(x) \mathbb{P}\left( \tilde{V}_\varphi(X_k, Y_k) \leq v \mid X_k \right) - \mathbb{P}\left( \tilde{V}_\varphi(X, Y) \leq v \mid X = x \right). \quad (37)$$

We now apply several results demonstrated later in this section:

- Applying Lemma 3 shows that

$$\mathbb{P}\left( 2 \sum_{k=1}^{m} \tilde{w}_k(x) \leq m \mathbb{E}[\tilde{w}_k(x)] \right) \leq \frac{4 \operatorname{Var}[\tilde{w}_k(x)]}{m \mathbb{E}[\tilde{w}_k(x)]}.$$

- Applying Theorem 8, for any $\gamma \in (0, 1)$, with probability at least $1 - \gamma$, it holds that

$$\sup_{v \in \mathbb{R}} \left\{ \sum_{k=1}^{m} \tilde{w}_k(x) \left\{ \mathbb{1}_{\tilde{V}_{\varphi,k} \leq v} - F_{\tilde{V}_\varphi | X}(v \mid X_k) \right\} \right\} < \sqrt{m \|K_{h_X}\|_\infty \log(1/\gamma)}.$$

- Applying Lemma 7, for any $\gamma \in (0, 1)$, with probability at least $1 - \gamma$, it follows that

$$\sup_{v \in \mathbb{R}} \left| \sum_{k=1}^{m} \tilde{w}_k(x) \left\{ F_{\tilde{V}_\varphi | X}(v \mid X_k) - F_{\tilde{V}_\varphi | X}(v \mid x) \right\} \right| \leq m D_{h_X}(x) + \sup_{t \in \mathbb{R}_+} \{ M t K_1(t) \} \sqrt{\frac{m \log(1/\gamma)}{2}},$$

where $D_{h_X}(x)$ is defined in (46).

Lastly, set $\gamma = m^{-1}$ and remark that $\|K_{h_X}\|_\infty = h_X^{-1} \|K_1\|_\infty$. Combining all the above bullet points with (37) implies, with probability at most $\frac{2}{m} + \frac{4 \operatorname{Var}[\tilde{w}_k(x)]}{m \mathbb{E}[\tilde{w}_k(x)]}$, that

$$\sup_{v \in \mathbb{R}} \left| \sum_{k=1}^{m} \tilde{w}_k(x) \mathbb{1}_{\tilde{V}_{\varphi,k} \leq v} - F_{\tilde{V}_\varphi | X}(v \mid x) \right| \geq \left( \sqrt{\frac{2 \|K_1\|_\infty}{h_X}} + \sup_{t \in \mathbb{R}_+} \{ M t K_1(t) \} \right) \sqrt{\frac{2 \log m}{m \mathbb{E}[\tilde{w}_k(x)]^2}} + \frac{2 D_{h_X}(x)}{\mathbb{E} \tilde{w}_k(x)}.$$

$\square$

**Corollary 1.** *If **H3** holds, then, it holds that*

$$\mathbb{P}\left( |\epsilon_\tau(x)| \geq \left( \sqrt{2 \|K_{h_X}\|_\infty} + \sup_{t \in \mathbb{R}_+} \{ M t K_1(t) \} \right) \sqrt{\frac{2 \log m}{m \mathbb{E}[\tilde{w}_k(x)]^2}} + \frac{2 D_{h_X}(x)}{\mathbb{E} \tilde{w}_k(x)} \right) \leq \frac{2 + 4 \mathbb{E}[\tilde{w}_k(x)]^{-1} \operatorname{Var}[\tilde{w}_k(x)]}{m}, \quad (38)$$

*where $D_{h_X}(x)$ is defined in (46), and $\lim_{h_X \to 0} D_{h_X}(x) = 0$.*

*Proof.* For $x \in \mathcal{X}$, since $\hat{F}_{\tilde{V}_\varphi | X = x}$ is continuous, applying Lemma 2 with Theorem 7 implies that (38) holds. Moreover, a calculation shows that

$$\limsup_{h_X \to 0} D_{h_X}(x) \leq \|F_X(\cdot, x)\|_\infty \int_0^\infty t^{d-1} K_1(t) \mathrm{d}t \times \limsup_{h_X \to 0} \left\{ h_X^{d-1} \right\}.$$

Finally, by **H3** we know that $\|F_X(\cdot, x)\|_\infty < \infty$ and $\int_{\mathbb{R}_+} t^{d-1} K_1(t) \mathrm{d}t < \infty$. Therefore, it follows that $\limsup_{h_X \to 0} D_{h_X}(x) = 0$. $\square$

The next result shows that $\sum_{k=1}^{m} \tilde{w}_k(x)$ concentrates around its mean with high probability.

**Lemma 3.** *If $\mathbb{E}[\tilde{w}_k(x)^2] < \infty$, then*

$$\mathbb{P}\left(2\sum_{k=1}^{m} \tilde{w}_k(x) \leq m\mathbb{E}[\tilde{w}_k(x)]\right) \leq \frac{4\operatorname{Var}[\tilde{w}_k(x)]}{m\mathbb{E}[\tilde{w}_k(x)]}.$$

*Proof.* Since the random variables $X_1, \ldots, X_m$ are i.i.d., using the Bienaymé-Tchebychev inequality, we obtain

$$\mathbb{P}\left(2\sum_{k=1}^{m} \tilde{w}_k(x) \leq m\mathbb{E}[\tilde{w}_k(x)]\right) \leq \frac{4\operatorname{Var}\left(\sum_{k=1}^{m} \tilde{w}_k(x)\right)}{\left(\sum_{k=1}^{m} \mathbb{E}\tilde{w}_k(x)\right)^2} = \frac{4\operatorname{Var}[\tilde{w}_k(x)]}{m\mathbb{E}[\tilde{w}_k(x)]}.$$

$\square$

### B.7.1. STEP 1: INTERMEDIATE RESULTS FOR THEOREM 7

For any $k \in [m]$ and $v \in \mathbb{R}$, let's recall that $\tilde{w}_k(x) = K_{h_X}(\|x - X_k\|)$ and let's define

$$G(v) = \sum_{k=1}^{m} \tilde{w}_k(x) \left\{ \mathbb{1}_{\tilde{V}_{\varphi,k} \leq v} - \mathbb{P}\left(\tilde{V}_{\varphi,k} \leq v \mid X_k\right) \right\}. \tag{39}$$

**Theorem 8.** *Let $x \in \mathcal{X}$ and $\gamma \in (0, 1)$. With probability at least $1 - \gamma$, the following inequality holds*

$$\sup_{v \in \mathbb{R}} \left\{ \sum_{k=1}^{m} \tilde{w}_k(x) \left\{ \mathbb{1}_{\tilde{V}_{\varphi,k} \leq v} - F_{\tilde{V}_\varphi|X}(v \mid X_k) \right\} \right\} < \sqrt{m\|K_{h_X}\|_\infty \log(1/\gamma)}.$$

*Proof.* Let $\theta > 0$, and denote by $\{\epsilon_k\}_{k \in [m]}$ a sequence of i.i.d. Rademacher random variables. The independence of $\{\tilde{w}_k(x)\}_{k \in [m]}$ implies that

$$\prod_{k=1}^{m} \mathbb{E}\left[\cosh(\theta\tilde{w}_k(x))\right] = \prod_{k=1}^{m} \left(2^{-1}\mathbb{E}\left[\exp(\theta\tilde{w}_k(x))\right] + 2^{-1}\mathbb{E}\left[\exp(-\theta\tilde{w}_k(x))\right]\right) = \prod_{k=1}^{m} \mathbb{E}\left[\exp(\theta\epsilon_k\tilde{w}_k(x))\right].$$

For all $x \in \mathbb{R}$, note that $\cosh(x) \leq \exp(x^2/2)$. Thus, we deduce that

$$\mathbb{E}\left[\exp(\theta\epsilon_k\tilde{w}_k(x))\right] \leq \exp\left(2^{-1}\theta^2\tilde{w}_k^2(x)\right).$$

Hence, the previous lines yields that

$$\prod_{k=1}^{m} \mathbb{E}\left[\cosh(\theta\tilde{w}_k(x))\right] \leq \exp\left(2^{-1}m\theta^2\|K_{h_X}\|_\infty^2\right). \tag{40}$$

Set $\Delta > 0$, applying Lemma 4 with $G$ defined in (39) gives

$$\mathbb{P}\left(\sup_{v \in \mathbb{R}}\{G(v)\} \geq \Delta\right) \leq 2\inf_{\theta > 0}\left\{e^{-\theta\Delta}\prod_{k=1}^{m} \mathbb{E}\left[\cosh(\theta\tilde{w}_k(x))\right]\right\}. \tag{41}$$

Now, consider the specific choice of $\theta_m$ given by

$$\theta_m = \frac{\Delta}{m\|K_{h_X}\|_\infty}.$$

Combining (40) with the expression of $\theta_m$, it follows that

$$\inf_{\theta > 0}\left\{e^{-\theta\Delta}\prod_{k=1}^{m} \mathbb{E}\left[\cosh(\theta\tilde{w}_k(x))\right]\right\} \leq \exp\left(-\frac{\Delta^2}{m\|K_{h_X}\|_\infty}\right).$$

Therefore, combining (41) with the previous inequality implies that

$$\mathbb{P}\left(\sup_{v \in \mathbb{R}}\{G(v)\} \geq \Delta\right) \leq \exp\left(-\frac{\Delta^2}{m\|K_{h_X}\|_\infty}\right). \tag{42}$$

For any $\gamma \in (0, 1)$, setting $\Delta = \sqrt{m\|K_{h_X}\|_\infty \log(1/\gamma)}$ gives

$$\mathbb{P}\left(\sup_{v \in \mathbb{R}}\{G(v)\} < \sqrt{m\|K_{h_X}\|_\infty \log(1/\gamma)}\right) \geq 1 - \gamma. \tag{43}$$

$\square$

The following statement controls $\mathbb{P}(\sup_{v \in \mathbb{R}}\{G(v)\} \geq \epsilon)$. Its proof is similar to the extension of the Dvoretzky–Kiefer–Wolfowitz inequality provided in Appendix B of (Plassier et al., 2024).

**Lemma 4.** *For any $\Delta > 0$, the following inequality holds*

$$\mathbb{P}\left(\sup_{v \in \mathbb{R}}\{G(v)\} \geq \Delta\right) \leq 2\inf_{\theta > 0}\left\{e^{-\theta\Delta}\prod_{k=1}^{m}\mathbb{E}\left[\cosh\left(\theta\tilde{w}_k(x)\right)\right]\right\},$$

*where $G$ is defined in* (39).

*Proof.* First, for any $\theta > 0$, applying Markov's inequality gives

$$\mathbb{P}\left(\sup_{v \in \mathbb{R}}\{G(v)\} \geq \Delta\right) \leq e^{-\theta\Delta}\mathbb{E}\left[\exp\left(\theta\sup_{v \in \mathbb{R}}\{G(v)\}\right)\right]. \tag{44}$$

Moreover, Lemma 5 shows that

$$\mathbb{E}\left[\exp\left(\theta\sup_{v \in \mathbb{R}}\{G(v)\}\right)\right] \leq 2\prod_{k=1}^{m}\mathbb{E}\left[\cosh\left(\theta\tilde{w}_k(x)\right)\right].$$

Plugging the previous inequality into (44), and minimizing the resulting expression with respect to $\theta$ yields:

$$\mathbb{P}\left(\sup_{v \in \mathbb{R}}\{G(v)\} \geq \Delta\right) \leq 2\inf_{\theta > 0}\left\{e^{-\theta\Delta}\prod_{k=1}^{m}\mathbb{E}\left[\cosh\left(\theta\tilde{w}_k(x)\right)\right]\right\}.$$

$\square$

**Lemma 5.** *Let $\theta > 0$, we have*

$$\mathbb{E}\left[\exp\left(\theta\sup_{v \in \mathbb{R}}\{G(v)\}\right)\right] \leq 2\prod_{k=1}^{m}\mathbb{E}\left[\cosh\left(\theta\tilde{w}_k(x)\right)\right].$$

*Proof.* Let $\theta > 0$ be fixed, since $t \mapsto e^{\theta t}$ is continuous and increasing, the supremum can be inverted with the exponential:

$$\mathbb{E}\left[\exp\left(\theta\sup_{v \in \mathbb{R}}\{G(v)\}\right)\right] = \mathbb{E}\left[\sup_{v \in \mathbb{R}}\exp\left(\theta G(v)\right)\right].$$

For any $k \in [m]$, consider $\tilde{Y}_k$ an independent copy of the random variable $Y_k$, and denote $\bar{V}_{\varphi,k} = \tilde{V}_\varphi(X_k, \tilde{Y}_k)$. The linearity of the expectation gives

$$\sum_{k=1}^{m}\tilde{w}_k(x)\left(\mathbb{1}_{\tilde{V}_{\varphi,k} \leq v} - \mathbb{E}\left[\mathbb{1}_{\tilde{V}_{\varphi,k} \leq v} \mid X_k\right]\right) = \mathbb{E}\left[\sum_{k=1}^{m}\tilde{w}_k(x)\left(\mathbb{1}_{\tilde{V}_{\varphi,k} \leq v} - \mathbb{1}_{\bar{V}_{\varphi,k} \leq v}\right) \;\Big|\; \{X_k, Y_k\}_{k=1}^{m}\right].$$

Therefore, the Jensen's inequality implies

$$\mathbb{E}\left[\exp\left(\theta\sup_{v\in\mathbb{R}}\{G(v)\}\right)\right] = \mathbb{E}\left[\sup_{v\in\mathbb{R}}\exp\left(\theta\mathbb{E}\left[\sum_{k=1}^m \tilde{w}_k(x)\left\{\mathbb{1}_{\tilde{V}_{\varphi,k}\leq v} - \mathbb{1}_{\bar{V}_{\varphi,k}\leq v}\right\} \,\Big|\, \{X_k, Y_k\}_{k=1}^m\right]\right)\right]$$

$$\leq \mathbb{E}\left[\sup_{v\in\mathbb{R}}\exp\left(\theta\sum_{k=1}^m \tilde{w}_k(x)\left\{\mathbb{1}_{\tilde{V}_{\varphi,k}\leq v} - \mathbb{1}_{\bar{V}_{\varphi,k}\leq v}\right\}\right)\right].$$

Let $\{\epsilon_k\}_{k\in[m]}$ be i.i.d. random Rademacher variables independent of $\{(X_k, Y_k, \tilde{Y}_k)\}_{k=1}^m$. Since $\mathbb{1}_{\tilde{V}_{\varphi,k}\leq v} - \mathbb{1}_{\bar{V}_{\varphi,k}\leq v}$ is symmetric, we have

$$\mathbb{E}\left[\sup_{v\in\mathbb{R}}\exp\left(\theta\sum_{k=1}^m \tilde{w}_k(x)\left\{\mathbb{1}_{\tilde{V}_{\varphi,k}\leq v} - \mathbb{1}_{\bar{V}_{\varphi,k}\leq v}\right\}\right)\right] = \mathbb{E}\left[\sup_{v\in\mathbb{R}}\exp\left(\theta\sum_{k=1}^m \epsilon_k\tilde{w}_k(x)\left\{\mathbb{1}_{\tilde{V}_{\varphi,k}\leq v} - \mathbb{1}_{\bar{V}_{\varphi,k}\leq v}\right\}\right)\right].$$

Using the Cauchy-Schwarz's inequality, we deduce that

$$\mathbb{E}\left[\exp\left(\theta\sup_{v\in\mathbb{R}}\{G(v)\}\right)\right] \leq \mathbb{E}\left[\sup_{v\in\mathbb{R}}\exp\left(2\theta\sum_{k=1}^m \epsilon_k\tilde{w}_k(x)\mathbb{1}_{\tilde{V}_{\varphi,k}\leq v}\right)\right].$$

Given the random variables $\{\tilde{V}_{\varphi,k}\}_{k=1}^m$, denote by $\sigma$ the permutation of $[m]$ such that $\tilde{V}_{\varphi,\sigma(1)} \leq \cdots \leq \tilde{V}_{\varphi,\sigma(m)}$. In particular, it holds

$$\sum_{k=1}^m \epsilon_k\tilde{w}_k(x)\mathbb{1}_{\tilde{V}_{\varphi,k}\leq v} = \begin{cases} 0 & \text{if } v < \tilde{V}_{\varphi,\sigma(1)} \\ \sum_{j=1}^i \epsilon_{\sigma(j)}\tilde{w}_{\sigma(j)}(x) & \text{if } \tilde{V}_{\varphi,\sigma(i)} \leq v < \tilde{V}_{\varphi,\sigma(i+1)} \\ \sum_{j=1}^m \epsilon_{\sigma(j)}\tilde{w}_{\sigma(j)}(x) & \text{if } v \geq \tilde{V}_{\varphi,\sigma(m)} \end{cases}.$$

Thus, can rewrite the supremum as

$$\sup_{v\in\mathbb{R}}\exp\left(2\theta\sum_{k=1}^m \epsilon_k\tilde{w}_k(x)\mathbb{1}_{\tilde{V}_{\varphi,k}\leq v}\right) \leq \sup_{0\leq i\leq n}\exp\left(2\theta\sum_{j=1}^i \epsilon_{\sigma(j)}\tilde{w}_{\sigma(j)}(x)\right).$$

Applying Lemma 6, we finally obtain that

$$\mathbb{E}\left[\sup_{v\in\mathbb{R}}\exp\left(2\theta\sum_{k=1}^m \epsilon_k\tilde{w}_k(x)\mathbb{1}_{\tilde{V}_{\varphi,k}\leq v}\right) \,\Big|\, \{X_k, Y_k\}_{k=1}^m\right]$$

$$\leq \mathbb{E}\left[\sup_{0\leq i\leq m}\exp\left(2\theta\sum_{j=1}^i \epsilon_{\sigma(j)}\tilde{w}_{\sigma(j)}(x)\right) \,\Big|\, \{X_k, Y_k\}_{k=1}^m\right] \leq 2\prod_{k=1}^m \cosh\left(\theta\tilde{w}_k(x)\right).$$

$\square$

**Lemma 6.** *Let $\{\epsilon_i\}_{i\in[n]}$ be i.i.d Rademacher random variables taking values in $\{-1, 1\}$, then for any $\theta > 0$ and $\{p_j\}_{j\in[m]} \in \mathbb{R}^m$, we have*

$$\mathbb{E}\left[\exp\left(\theta\sup_{0\leq i\leq m}\sum_{j=1}^i p_j\epsilon_j\right)\right] \leq 2\prod_{k=1}^m \cosh\left(\theta p_k\right).$$

*By convention, we consider $\sum_{j=1}^0 p_j\epsilon_j = 0$.*

### B.7.2. STEP 2: INTERMEDIATE RESULTS FOR THEOREM 7

For all $x \in \mathcal{X}$ and $v \in \mathbb{R}$, define the conditional cumulative density function $F_{\tilde{V}_\varphi|X}(v \mid x)$ as

$$F_{\tilde{V}_\varphi|X}(v \mid x) = \mathbb{P}\left(\tilde{V}_\varphi(X, Y) \leq v \mid X = x\right).$$

Moreover, recall that we denote by $f_X$ the density with respect to the Lebesgue measure of the random variable $X$. Using the spherical coordinates, we write by $\tilde{x}_{t,\theta} = (t\cos\theta_1, t\sin\theta_1\cos\theta_2, \ldots, t\sin\theta_1 \cdots \sin\theta_{d-1})$ the coordinate of $\tilde{x} \in \mathcal{X}$, where $\|\tilde{x}\| = t$. Additionally, we define

$$F_X(t, x) = \int_{[0,\pi]^{d-2}\times[0,2\pi)} f_X(x - \tilde{x}_{t,\theta}) \prod_{i=1}^{d-2} \sin(\theta_i)^{d-1-i} \mathrm{d}\theta_1 \cdots \mathrm{d}\theta_{d-1}. \tag{45}$$

Note that

$$\int_{\mathbb{R}_+} t^{d-1} F_X(t, x)\mathrm{d}t = \int_{\mathcal{X}} f_X(x - \tilde{x})\mathrm{d}\tilde{x} = 1.$$

Under **H**3 the cumulative density function $x \mapsto F_{\tilde{V}_\varphi|X}(v \mid x)$ is M-Lipschitz. In this case, for any $h_X > 0$, let's consider

$$D_{h_X}(x) = h_X^{d-1}\|F_X(\cdot, x)\|_\infty \int_0^\infty t^{d-1} K_1(t)\mathrm{d}t. \tag{46}$$

**Lemma 7.** *Assume that **H**3 holds and let $\gamma \in (0, 1)$. With probability at least $1 - \gamma$, it holds*

$$\sup_{v\in\mathbb{R}} \left| \sum_{k=1}^m \tilde{w}_k(x)\left\{ F_{\tilde{V}_\varphi|X}(v \mid X_k) - F_{\tilde{V}_\varphi|X}(v \mid x)\right\}\right| \leq mD_{h_X}(x) + \sup_{t\in\mathbb{R}_+}\{MtK_1(t)\}\sqrt{\frac{m\log(1/\gamma)}{2}}.$$

*Proof.* First of all, using **H**3 implies that

$$\sup_{v\in\mathbb{R}} \left| \sum_{k=1}^m \tilde{w}_k(x)\left\{ F_{\tilde{V}_\varphi|X}(v \mid X_k) - F_{\tilde{V}_\varphi|X}(v \mid x)\right\}\right| \leq \sum_{k=1}^m \tilde{w}_k(x)\min\left\{1, M\|x - X_k\|\right\}.$$

For every $k \in [m]$, let's consider $Z_k = \tilde{w}_k(x)\min\{1, M\|x - X_k\|\}$. Since $\tilde{w}_k = K_{h_X}(\|x - X_k\|)$, we have

$$Z_k \leq \max\left(\sup_{0\leq Mt\leq 1}\{MtK_{h_X}(t)\}, \sup_{Mt>1}\{K_{h_X}(t)\}\right). \tag{47}$$

By calculation, we get

$$\sup_{0\leq Mt\leq 1}\{MtK_{h_X}(t)\} = M\sup_{0\leq Mt\leq 1}\left\{\frac{t}{h_X}K_1\left(\frac{t}{h_X}\right)\right\} = M\sup_{0\leq t\leq (h_X M)^{-1}}\{tK_1(t)\}. \tag{48}$$

We also have

$$\sup_{Mt>1}\{K_{h_X}(t)\} = \sup_{Mt>1}\left\{\frac{1}{h_X}K_1\left(\frac{t}{h_X}\right)\right\} \leq M\sup_{Mt>1}\left\{\frac{t}{h_X}K_1\left(\frac{t}{h_X}\right)\right\} = M\sup_{t>(h_X M)^{-1}}\{tK_1(t)\}. \tag{49}$$

Thus, combining (47)-(48) with (49) yields

$$0 \leq Z_k \leq M\sup_{t\in\mathbb{R}_+}\{tK_1(t)\}.$$

Applying Hoeffding's inequality, for any $t > 0$, it follows

$$\mathbb{P}\left(\sum_{k=1}^m (Z_k - \mathbb{E}Z_k) \geq t - m\mathbb{E}Z_1\right) \leq \exp\left(-\frac{2(t - m\mathbb{E}Z_1)^2}{m\sup_{t\in\mathbb{R}_+}\{MtK_1(t)\}^2}\right). \tag{50}$$

Let $\gamma \in (0, 1)$ and set:

$$t_\gamma = m\mathbb{E}Z_1 + \sup_{t\in\mathbb{R}_+}\{MtK_1(t)\}\sqrt{\frac{m\log(1/\gamma)}{2}}.$$

Using (50), it holds that

$$\mathbb{P}\left(\sum_{k=1}^m (Z_k - \mathbb{E}Z_k) \geq t_\gamma - m\mathbb{E}Z_1\right) \leq \gamma.$$

We will now bound $t_\gamma$. To do this, we will control $\mathbb{E}Z_1$:

$$\mathbb{E}\left[\tilde{w}_k(x)\min\{1, \mathrm{M}\|x - X_k\|\}\right] = \int_{\tilde{x} \in \mathcal{X}} (\mathrm{M}\|\tilde{x}\| \wedge 1)\, K_{h_X}(\|\tilde{x}\|) f_X(x - \tilde{x}) \mathrm{d}\tilde{x}. \tag{51}$$

Using the spherical coordinates, a change of variables gives

$$\int_{\tilde{x} \in \mathcal{X}} (\mathrm{M}\|\tilde{x}\| \wedge 1)\, K_{h_X}(\|\tilde{x}\|) f_X(x - \tilde{x}) \mathrm{d}\tilde{x} = \int_{t=0}^{\infty} t^{d-1} (\mathrm{M}t \wedge 1) K_{h_X}(t) F_X(t, x) \mathrm{d}t,$$

where $F_X(t, x)$ is given in (45). Therefore, it immediately follows that

$$\int_{\tilde{x} \in \mathcal{X}} (\mathrm{M}\|\tilde{x}\| \wedge 1)\, K_{h_X}(\|\tilde{x}\|) f_X(x - \tilde{x}) \mathrm{d}\tilde{x} \leq h_X^{d-1} \int_{t=0}^{\infty} t^{d-1} K_1(t) F_X(h_X t, x) \mathrm{d}t.$$

Plugging the previous bound in (51) shows that

$$\mathbb{E}Z_1 \leq D_{h_X}(x), \qquad \text{where } D_{h_X}(x) \text{ is provided in (46)}.$$

$\square$

## B.8. Asymptotic conditional validity

**Theorem 9.** *Assume that **H1**-**H2**-**H3** hold, and let $m$ be of the same order as $n$. If $F_{\tilde{V}}(\varphi) \notin \{0, 1\}$ and for every $x \in \mathcal{X}$, $F_{\tilde{V}|X=x} \circ F_{\tilde{V}}^{-1}$ is Lipschitz and $f_X$ is continuous, then, for $\alpha \in [\{n + 1\}^{-1}, 1)$ and $\rho > 0$, it follows*

$$\lim_{h_X \to 0} \lim_{n \to \infty} \mathbb{P}\left(|\mathbb{P}\left(Y \in \mathcal{C}_\alpha(X)\,|\,X\right) - 1 + \alpha| \leq \rho\right) = 1.$$

*Proof.* First, let's fix $\alpha \in [\{n + 1\}^{-1}, 1)$ and $\rho > 0$. Our proof is based on the following set:

$$A_n = \left\{x \in \mathcal{X}\colon F_{\tilde{V}|X=x} \circ F_{\tilde{V}}^{-1} \text{ is } 4^{-1}\rho[F_{\tilde{V}}(\varphi)]^{-n} \wedge [1 - F_{\tilde{V}}(\varphi)]^{n+1}\text{-Lipschitz}\right\}.$$

This set contains every point $x \in \mathcal{X}$ whose Lipschitz constant of $F_{\tilde{V}|X=x} \circ F_{\tilde{V}}^{-1}$ is smaller than a certain threshold which tends to $\infty$ as $n \to \infty$. Let's also define the two following sets

$$B_{m,h_X} = \left\{x \in \mathcal{X}\colon \frac{\sqrt{2\|K_{h_X}\|_\infty} + \sup_{t \in \mathbb{R}_+}\{MtK_1(t)\}}{C_{h_X}(x)}\sqrt{\frac{2\log m}{m}} + \frac{2D_{h_X}(x)}{C_{h_X}(x)} \leq \frac{\rho}{2}\right\},$$

$$B_{\infty,h_X} = \left\{x \in \mathcal{X}\colon \frac{2D_{h_X}(x)}{C_{h_X}(x)} \leq \frac{\rho}{2}\right\}.$$

Lastly, for all $r > 0$, consider

$$E_{m,h_X} = \left\{x \in \mathcal{X}\colon 2 + 4C_{h_X}(x)^{-1}\,\mathrm{Var}[K_{h_X}(\|x - X\|)] \leq mr\right\},$$
$$G = \{x \in \mathcal{X}\colon f_X(x) \geq r\}.$$

Using basic computations, we obtain the following line

$$\mathbb{P}\left(|\mathbb{P}\left(Y \in \mathcal{C}_\alpha(X)\,|\,X\right) - 1 + \alpha| > \rho\right) \leq \mathbb{P}\left(X \notin A_n \cap B_{m,h_X} \cap E_{m,h_X}; X \in G\right)$$
$$+ \mathbb{P}\left(X \notin G\right) + \mathbb{P}\left(|\mathbb{P}\left(Y \in \mathcal{C}_\alpha(X)\,|\,X\right) - 1 + \alpha| > \rho; X \in A_n \cap B_{m,h_X} \cap E_{m,h_X}\right). \tag{52}$$

Since $m$ is of the same order as $n$, which means that $0 < \liminf m/n \leq \limsup m/n < \infty$, it holds

$$\mathbb{1}_{X \notin A_n \cap B_{m,h_X} \cap E_{m,h_X}}\,\mathbb{1}_{X \in G} \xrightarrow[n \to \infty]{} \mathbb{1}_{X \notin B_{\infty,h_X}}\,\mathbb{1}_{X \in G}.$$

Moreover, since $K_{h_X}$ is an approximate identity and $f_X$ is continuous and bounded, we have $\lim_{h_X \to 0} C_{h_X}(x) = f_X(x)$. As stated in Proposition 1, it also holds that $\lim_{h_X \to 0} D_{h_X}(x) = 0$. Therefore, it follows

$$\mathbb{1}_{X \notin B_{\infty,h_X}}\,\mathbb{1}_{X \in G} \xrightarrow[h_X \to 0]{} 0.$$

Using the dominated convergence theorem, it yields that

$$\limsup_{h_X \to 0} \limsup_{n \to \infty} \mathbb{P}\left(X \notin A_n \cap B_{m,h_X} \cap E_{m,h_X}; X \in G\right) = 0. \tag{53}$$

Given a realization $x \in \mathcal{X}$, denoting by $\mathrm{L}_x$ the Lipschitz constant of $F_{\tilde{V}|X=x} \circ F_{\tilde{V}}^{-1}$, the application of Theorem 2 shows that

$$\mathbb{P}\left(\left|\mathbb{P}\left(Y \in \mathcal{C}_\alpha(X) \mid X\right) - 1 + \alpha\right| > \rho; X \in A_n \cap B_{m,h_X} \cap E_{m,h_X}\right)$$
$$\leq \mathbb{P}\left(\rho < |\epsilon_\tau(X)| + 2\mathrm{L}_X \times [F_{\tilde{V}}(\varphi)]^{n+1} \vee [1 - F_{\tilde{V}}(\varphi)]^{n+1}; X \in A_n \cap B_{m,h_X} \cap E_{m,h_X}\right).$$

Since $X \in A_n$, we deduce that $\mathrm{L}_X \leq 4^{-1}\rho[F_{\tilde{V}}(\varphi)]^{-n} \wedge [1 - F_{\tilde{V}}(\varphi)]^{n+1}$, and thus it yields that $2\mathrm{L}_X \times [F_{\tilde{V}}(\varphi)]^{n+1} \vee [1 - F_{\tilde{V}}(\varphi)]^{n+1} \leq 2^{-1}\rho$. Therefore, it follows

$$\mathbb{P}\left(\rho < |\epsilon_\tau(X)| + 2\mathrm{L}_X \times [F_{\tilde{V}}(\varphi)]^{n+1} \vee [1 - F_{\tilde{V}}(\varphi)]^{n+1}; X \in A_n \cap B_{m,h_X} \cap E_{m,h_X}\right)$$
$$\leq \mathbb{P}\left(2^{-1}\rho < |\epsilon_\tau(X)|; X \in A_n \cap B_{m,h_X} \cap E_{m,h_X}\right).$$

Since $x \in B_{m,h_X} \cap E_{m,h_X}$, applying Proposition 1 gives that

$$\mathbb{P}\left(2^{-1}\rho < |\epsilon_\tau(X)|; X \in A_n \cap B_{m,h_X} \cap E_{m,h_X}\right) \leq r.$$

Equation (52) implies

$$\mathbb{P}\left(\left|\mathbb{P}\left(Y \in \mathcal{C}_\alpha(X) \mid X\right) - 1 + \alpha\right| > \rho\right) \leq \mathbb{P}\left(X \notin A_n \cap B_{m,h_X} \cap E_{m,h_X}\right) + \mathbb{P}\left(X \notin G\right) + r.$$

Lastly, (53) combined with the previous inequality shows

$$\limsup_{h_X \to 0} \limsup_{n \to \infty} \mathbb{P}\left(\left|\mathbb{P}\left(Y \in \mathcal{C}_\alpha(X) \mid X\right) - 1 + \alpha\right| > \rho\right) \leq \mathbb{E}\left[\mathbb{1}_{f_X(X) < r}\right] + r.$$

As $r$ is arbitrary fixed, from the dominated convergence theorem we can conclude that

$$\limsup_{h_X \to 0} \limsup_{n \to \infty} \mathbb{P}\left(\left|\mathbb{P}\left(Y \in \mathcal{C}_\alpha(X) \mid X\right) - 1 + \alpha\right| > \rho\right) = 0.$$

$\square$

# C. Details on Experimental Setup

This section aims to provide additional details on our experimental setup and implementation of the RCP algorithm.

**Models.** To facilitate fair comparison of different uncertainty estimation methods, we assume that the base prediction models are already trained. We focus on the regression problem and aim to construct prediction sets for these pre-trained models. All our models are based on a fully connected neural network of three hidden layers with 100 neurons in each layer and ReLU activations. We consider three types of base models with appropriate output layers and loss functions: the mean squared error for the *mean predictor*, the pinball loss for the *quantile predictor* or the negative log-likelihood loss for the *mixture predictor*. Training is performed with Adam optimizer.

Each dataset is split randomly into train, calibration, and test parts. We reserve 2048 points for calibration and the remaining data is split between 70% for training and 30% for testing. Each dataset is shuffled and split 10 times to replicate the experiment. This way we have 10 different models for each dataset and these models' prediction are used by every method that is tailored to the corresponding model type to estimate uncertainty. One fifth of the train dataset is reserved for early stopping.

**RCP$_{\mathrm{MLP}}$.** This variation reserves a part (50%) of the original calibration set to train a quantile regression model for the $(1 - \alpha)$-level quantile of the scores $V$. We again use a three hidden layers with 100 units per layers for that task. The remaining half of the calibration set forms the "proper calibration set" and is used to compute the conformal correction.

**RCP$_{\mathrm{local}}$.** The local quantile regression variant is similar to the previous one, so we use the same splitting of the available calibration data. Since only one bandwidth needs to be tuned, we use a simple grid search on a log-scale grid in the interval $[10^{-3}, 1]$.

**Datasets.** Table 6 presents characteristics of datasets from (Tsoumakas et al., 2011; Feldman et al., 2023; Wang et al., 2023), restricting our selection to those with at least two outputs and a total of 2000 instances. For data preprocessing, we follow the procedure of (Grinsztajn et al., 2022).

| Paper | Dataset | $n$ | $p$ | $d$ |
|---|---|---|---|---|
| Tsoumakas et al. (2011) | scm20d | 8966 | 60 | 16 |
| | rf1 | 9005 | 64 | 8 |
| | rf2 | 9005 | 64 | 8 |
| | scm1d | 9803 | 279 | 16 |
| Feldman et al. (2023) | meps_21 | 15656 | 137 | 2 |
| | meps_19 | 15785 | 137 | 2 |
| | meps_20 | 17541 | 137 | 2 |
| | house | 21613 | 14 | 2 |
| | bio | 45730 | 8 | 2 |
| | blog_data | 50000 | 55 | 2 |
| Wang et al. (2023) | taxi | 50000 | 4 | 2 |

Table 6: List of datasets with their characteristics.

