# OpenReview forum: "Rectifying Conformity Scores for Better Conditional Coverage"
_ICML.cc/2025/Conference — ICML 2025 poster_

### Official Review · Reviewer_1t9h · 2025-02-18

**Overall Recommendation:** 3

**Summary:**

The paper presents a novel method to achieve better conditional coverage in conformal prediction for single-output and multi-output regression. The central idea is to start from a classical nonconformity score, and adjust it to improve for conditional coverage. The adjustment is a factor that is obtained by estimating conditional quantiles using classical or local quantile regression.

The authors present theoretical results, claiming that their proposed method achieves the desired marginal and conditional coverage, provided that conditional quantiles are known.

The experiments on synthetic and real-world data intend to show that the proposed method works well in practice. On real-world datasets improvements in conditional coverage are observed compared to four baseline methods.

**Claims And Evidence:**

The main goal of the paper is to present a new method that improves on conditional coverage.

I believe that the presented method is novel, but it is a pity that the authors don't discuss the limitations of their approach.

I enjoyed reading the theoretical discussion in Section 3, but I am somewhat less convinced of the practical implementation in Section 4. Conformalized quantile regression has been proposed in the literature as a tool to improve the quantiles obtained by quantile regression, so that better conditional coverage is obtained. Here the authors are reasoning the other way around: they are using quantile regression to improve what conformal prediction is doing wrong... So, others have claimed that quantile regression is not good in obtaining quantiles for regression problems that are strongly heteroskedastic, while here is claimed that quantile regression is the solution. I believe that this deserves more discussion...

In light of this, the experiments with synthetic data are also not convincing. It is obvious that conditional coverage will be obtained if access to the ground-truth conditional distribution is assumed. I would have liked to see on synthetic data how the method performs when the quantiles need to be estimated using quantile regression. The considered toy problem is strongly heteroskedastic, so I assume that estimating the quantiles is far from trivial, despite the one-dimensionality of the problem in feature space.

Apart from the connection with conformalized quantile regression, I believe that the proposed method is also closely related to the "normalized" conformal prediction literature. This literature is not discussed in the related work section, but normalized nonconformity scores have a very similar idea in mind. For regression, the standard nonconformity score based on absolute residuals is divided by (an estimate) of the variance, see e.g.:

H. Papadopoulos, A. Gammerman, and V. Vovk. Normalized nonconformity measures for regression conformal
prediction. In Proceedings of the IASTED International Conference on Artificial Intelligence and Applications
(AIA 2008), pages 64–69, 2008.

U. Johansson, H. Boström, and T. Löfström. Investigating normalized conformal regressors. In 2021 IEEE
Symposium Series on Computational Intelligence (SSCI), pages 1–8. IEEE, 2021.

N. Dewolf, B. De Baets ,and W. Waegeman, Conditional validity of heteroskedastic conformal regression. Arxiv 2023.

Under certain assumptions, exact conditional coverage is obtained, similar to the reasoning of the authors. So, I think that this literature should be discussed. Moreover, normalized conformal prediction would also be the most obvious baseline in the experiments. Conformalized quantile regression would also be an obvious baseline. I did not understand the reasoning of the authors for the baselines they chose.

**Essential References Not Discussed:**

See above.

**Experimental Designs Or Analyses:**

See above.

**Methods And Evaluation Criteria:**

See previous section.

**Other Comments Or Suggestions:**

None.

**Other Strengths And Weaknesses:**

Strengths:
- The paper is very well written (the authors give evidence of a solid math background, which is appreciated)
- The proposed method is novel
- The authors present non-trivial theoretical results

Weaknesses:
- The limitations are not discussed
- Related work is missing
- The experiments are a bit underwhelming.

**Questions For Authors:**

I invite the authors to give feedback on my comments.

**Relation To Broader Scientific Literature:**

See above.

**Theoretical Claims:**

The theoretical claims make sense to me. I did not check the proofs in detail, but the claims are pretty straight-forward, so I don't see issues.

I don't understand why assumption H2 is needed. This is a very general assumption that is always fulfilled in practice, isn't it? Perhaps this assumption can be simply omitted.

---

> ### Author Rebuttal · Authors · 2025-03-30
>
> We thank the reviewer for the thorough and constructive feedback, which helps improve our manuscript. Below, we address your valuable points:
>
> The **main limitations** of RCP can be summarized as follows:
> - The quality of the prediction regions heavily depends on the basic conformity score $V$. For example, if the underlying multi-output conformal method predicts hyperrectangular sets, RCP will also predict hyperrectangular sets.
> - The quality of the conditional coverage of the intervals crucially depends on the quality of the conditional quantile estimator $\hat{\tau}(x)$ of the conformity score. The marginal coverage guarantee is **always ensured**; however, the guarantee of conditional coverage clearly depends on how $\hat{\tau}(x)$ approaches $\tau_*(x)$, the "exact" quantile. A key contribution of our work is the explicit tracking of how errors in quantile estimation influence conditional coverage error.
>
> **Additional experiment on synthetic data.** Following your advice, we conducted an additional experiment on synthetic data to investigate the case of learned quantile estimate. We used a simple MLP trained on a dataset of varying sizes ranging from 100 to 500 points (note that the calibration dataset size in this experiment is equal to 500). The resulting plot can be found via https://pdfhost.io/v/ND3Dt3PahY_Additional_synthetic_data_experiment
> We see that the conditional coverage is not perfect, but RCP outperforms the standard CP already for a relatively small data size of 100 points.
>
> **Why assumption H2 is needed?** This assumption ensures that the score function is compatible with the adjustment function. Only valid $t$ values will be supplied to $f\_t(v)$ to ensure that H1 is satisfied. For example, if $f_t(v)=tv$ then $\hat{\tau}(x)$ has to be positive.
>
> **Connection with normalized conformity scores.**
> There is indeed a connection to prior works on normalized conformity scores. We will incorporate these references into the literature review of the camera-ready version (if accepted). Normalized Conformity Scores (NCS) all share the core idea of normalizing nonconformity scores by the predictive accuracy of the underlying model at new data points. This normalization aims to enhance the efficiency of conformal predictions by assigning wider prediction intervals to challenging instances and narrower intervals to easier ones, with the difficulty determined by the accuracy of the predictive model itself. However, these studies generally lack a detailed analysis of approximate conditional coverage, which distinguishes our work. Furthermore, we can recover the specific formulation of normalized nonconformity scores through an appropriate choice of the function $f_{\tau}(v)$. However, the criterion employed in our method to estimate $\hat{\tau}(x)$ fundamentally differs.
>
> We provide further details below.
> - U. Johansson et al. [1] and Papadopoulos et al. [2] investigate NCS methods, which enhance standard conformal prediction by dynamically adjusting prediction interval sizes according to instance difficulty. Normalization in their methods involves a parameter $\beta$, which balances the model's prediction error and the estimation of difficulty. However, these methods lack explicit theoretical guarantees. Their NCS can be represented within our framework through a specific choice of the function $f_{\tau}(v) = v/(\tau + \beta)$. Notably, the estimation approach employed in these papers uses least-squares regression on residuals, in contrast to the quantile regression approach adopted in RCP. The central goal in RCP is to construct a "rectified" conformity score that aligns conditional and unconditional quantiles at a target level--an objective distinct from NCS.
>
> - The paper by Dewolf et al. [3] provides an insightful summary of normalized conformal predictors, introducing the concept of a taxonomy function, which they assume to be discrete. In their own terms, the taxonomy function "divides the instance space based on an estimate of the uncertainty," for example, by partitioning the feature space through binning the (conditional) standard deviation. However, their analysis is restricted to an oracle setting, meaning that the theoretical developments rely on an exact, known normalizing function. Consequently, their work does not address the practical scenario in which this normalizing function must be estimated from data.
>
> [1] H. Papadopoulos, A. Gammerman, and V. Vovk. Normalized nonconformity measures for regression conformal prediction. In Proceedings of the IASTED International Conference on Artificial Intelligence and Applications (AIA 2008), pages 64–69, 2008.
>
> [2] U. Johansson, H. Boström, and T. Löfström. Investigating normalized conformal regressors. In 2021 IEEE Symposium Series on Computational Intelligence (SSCI), pages 1–8. IEEE, 2021.
>
> [3] N. Dewolf, B. De Baets ,and W. Waegeman, Conditional validity of heteroskedastic conformal regression. Arxiv 2023.

---

### Official Review · Reviewer_R3jf · 2025-03-11

**Overall Recommendation:** 2

**Summary:**

The paper considers the problem of producing conformal prediction sets with conditional guarantees. The idea is to rectify non-conformity scored by to use additional hold out data to fit a quantile regressor that is then applied to the non-conformity score. Marginal coverage guarantees are obtained using the remaining part of the data to run split cp on the rectified nc scores.

**Claims And Evidence:**

The paper claims to improve conditional coverage compared to existing schemes; however, as shown in the Appendix the proposed method has very similar performance to well established methods as CQR and it is not compared to methods such as those presented in  Conformal prediction with conditional guarantees by Gibbs, I., Cherian, J. J., and Candes, E. J., and Boosted Conformal Prediction Intervals by R Xie, R Barber, E Candes.

**Essential References Not Discussed:**

The idea of optimizing the non-conformity score for improved efficiency and conditional coverage is common in conformal training and CP length optimization. See:

Large language model validity via enhanced conformal prediction methods by John J. Cherian, Isaac Gibbs, Emmanuel J. Candès

Kiyani S, Pappas GJ, Hassani H. Length optimization in conformal prediction. Advances in Neural Information Processing Systems.

R Xie, R Barber, E Candes, Boosted Conformal Prediction Intervals

**Experimental Designs Or Analyses:**

Yes, experiments are correct. However I believe that other baselines (CQR) should be either moved to the main text and additional ones considered (such as those mentioned above based on boosting, conditional coverage and conformal training should be included.

**Methods And Evaluation Criteria:**

Yes, benchmarks make sense.

**Other Comments Or Suggestions:**

I have found problems following the methodology section given that at some point is set f_t(v)=\tilde f_v(t). Is this really necessary?

**Other Strengths And Weaknesses:**

I think the idea is simple, but that is not necessarily a weakness of the paper. However, I had trouble understanding the benefits and novelty of the proposed scheme compared to existing methods such as CQR, conformal training, and conditional coverage methods. This concern arises from the fact that these methods are either not benchmarked or, in the case of CQR, have similar or superior performance to the proposed one (e.g., in Figure 9, CQR has the same conditional coverage but a smaller volume?)

**Questions For Authors:**

What is exactly the technical novelty and benefits of the proposed method as compared to the above mentioned schemes? How do they empirically compare?

Update after rebuttal:  I have decided to update my score from reject to weak reject. There exists a fairly large number of relevant papers that are not discussed or benchmarked. This is a common concern among other reviewers; however, they did not consider it as serious as I did—hence my score upgrade.

**Relation To Broader Scientific Literature:**

I think the paper does a good job at highlighting similarities with existing literature; however I believe it misses on discussing conformal training methods.

**Theoretical Claims:**

The theoretical claims appear to correct as they follow from the properties of split cp which is applied to the rectified scores. What are the technical challenges and novelty that the current analysis brings?

---

> ### Author Rebuttal · Authors · 2025-03-28
>
> We acknowledge the critical feedback and aim to address the raised points. Below we clarify why **the undiscussed references, while indeed valuable, mostly address aspects different from the specific problem we focus on**. The following discussion highlights the distinctive advantages of our RCP method for constructing confidence sets in multivariate prediction, offering exact marginal coverage and strong theoretical guarantees for conditional coverage.
>
> **Discussion of the references provided in the review.**
> - Among the papers cited, Gibbs et al. (2023) is the one most directly related to our work, as it acts as a wrapper on given conformity scores. It has already been discussed, but additional comments and experiments are worthwhile. From a computational point of view, their method (called CPCG below) is very intensive, mainly because the wrapper uses a form of the "full conformal" idea, which requires solving an optimization problem at test time. Its strength lies mainly in finite-sample guarantees tailored explicitly to specific feature subsets and covariate shifts (see Theorem 2 for finite-dimensional class). This targeted approach suits group conditional coverage (Corollary 1) well. In contrast, RCP uses a split-conformal idea that estimates a quantile function on a special subset of the data, which greatly improves computational complexity. Also, for RCP, we provide a more generic theoretical result that is agnostic to a particular estimation method used (Theorem 4), and also specify it for the case of local quantile regression (Proposition 5). The proof of the results is not straightforward and requires the usage of non-trivial technical tools such as very recent extensions of DKW inequality (see Lemma 16).
>
> - Cherian et al. (2024) is a CP framework tailored for LLMs. The method is not easily applicable to conventional regression or standard multivariate classification scenarios, where more broadly effective methods like RCP are more natural.
>
> - Xie et al. (2024) refine conformity scores via gradient boosting to enhance conditional coverage while achieving exact marginal coverage in univariate prediction models. This method is constrained to scalar outcomes and lacks a natural extension to multidimensional settings. Further, the lack of analytically interpretable theoretical guarantees on conditional coverage and reliance on numerous hyperparameters and differentiable approximations complicates both theoretical understanding and practical implementation.
>
> - Kiyani’s (2024) CPL method combines conditional validity and optimized efficiency through constrained optimization, designed exclusively for univariate prediction intervals. While CPL provides exact coverage tailored for a specific class $\mathcal{F}$, it is fundamentally limited by its complexity. Specifically, the reliance on intricate optimization procedures limits broader practical application, especially for multivariate outputs. In contrast, RCP’s computational simplicity and broader versatility in handling multivariate scenarios position it distinctly ahead.
>
> Thus, among these methods, RCP clearly emerges as superior, particularly in multivariate predictive contexts. It balances precision in conditional coverage with practical simplicity, computational efficiency, and broader applicability.
>
> **Empirical comparison with CQR and CPCG.** We conducted an additional experiment to directly compare RCP and CPCG (see https://pdfhost.io/v/26gN4fUeS2_rebuttal-R3jf).
> We observe that CPCG and RCP give similar conditional coverage, while RCP is at least two orders of magnitude faster. Compared to CQR, it trains a quantile regression model directly on the whole training set, while RCP can be applied to any black-box model and does not require access to the training data or internal model structure. Figure 10 shows that RCP can match or outperform CQR in a multidimensional setting.
>
> **Technical novelty.**
> The **technical novelty** of RCP lies in its approach to enhancing conditional coverage through the concept of "rectifying" conformity scores. Unlike conventional methods requiring estimating the entire conditional distribution for multivariate predictions, RCP simplifies the problem by calculating only the conditional quantile of a univariate conformity score. This quantile estimation is a wrapper around classical methods explicitly tailored for multivariate prediction sets. Furthermore, RCP provides explicit theoretical lower bounds on conditional coverage, directly linking prediction accuracy to the quantile estimation approximation error. These results are based on careful and non-trivial analysis as discussed above. **While competitive methods exist for univariate predictions, such comparisons are irrelevant, as univariate scenarios are explicitly not our targeted application**. Thus, our proposed RCP method represents a **meaningful advancement**, combining clear theoretical foundations with practical efficiency for multivariate prediction tasks.

---

> > ### Comment · Reviewer_R3jf · 2025-04-07
> >
> > Thank you for having taken the time to address my comments!
> >
> > Regarding the relevance of Kiyani’s and Cherian’s work, it lies in the fact that their approaches optimize a parameterized non-conformity scoring function to improve efficiency and conditional coverage. In that sense, the idea of “rectifying” non-conformity scores by choosing within a family of parametric transformations, as RCP does, is related.
> >
> > However, I don’t understand why previous literature is completely disregarded, given that it is claimed to be limited to uni-dimensional target variables. Many of the mentioned alternatives operate directly on the non-conformity scoring function, making them independent of the target’s dimensionality. Once a non-conformity scoring function is established—even for multidimensional targets—the methods proposed by Kiyani and Cherian still apply. The same principle holds for simpler methods, such as variance-reduced (or normalized) non-conformity scores, where errors are simply divided by the average. The gap between unimodal and multimodal approaches isn’t just about replacing absolute values with norms, is it? Consider the work of Colombo "On training locally adaptive CP". Aren't all the proposed transformation applicable to this setting by simply the norm error $\lVert f(X)-Y\rVert$, instead of the absolute error?
> >
> > In my original reply, I mistakenly referred to Figure 10 as Figure 9. However, the concern remains. How do you conclude that RCP outperforms CQR based on this figure? RCP outperforms CQR in only four datasets and is outperformed by CQR in two. I would not conclude that one method consistently outperforms the other. The same applies to conditional coverage—there is no clear winner in that metric either.
> >
> > Thanks again for taking the time to write the rebuttal and provide the additional experiments.

---

> > > ### Author Response · Authors · 2025-04-08
> > >
> > > Dear reviewer R3jf,
> > >
> > > we thank you for your comments and address your remaining questions and concerns below.
> > >
> > > 1. We strongly disagree with the assertion that previous literature has been completely disregarded; on the contrary, our work makes extensive reference to relevant previous studies (among the 60+ references, more than 40 are recent - less than 10 years). The literature on conformal methods is extraordinarily vast, and conducting a concise state-of-the-art review inherently requires deliberate selection and prioritization. In this paper, we have intentionally focused on the most prevalent and widely adopted approaches, particularly those we evaluate directly through our benchmarks. We maintain that our selection contains no significant omissions. Nevertheless, we acknowledge the potential relevance of additional contributions and elaborate further on this point below. We will also do it in a final version of the paper.
> > >
> > > - The works by **Cherian and Kiyani** are indeed valuable contributions; however, their conditional coverage guarantees are restricted to specific classes of functions. Consequently, these methods do not achieve the same form of pointwise approximate conditional coverage that we establish in our approach. While it is appropriate to acknowledge these references within our literature review, a direct empirical comparison is not feasible due to fundamental differences in methodological setups and underlying assumptions.
> > >
> > > - The paper by **Colombo** is indeed highly relevant, and we agree that it is appropriate to explicitly include it in our discussion. We thank the reviewer for drawing our attention to this reference. However, we emphasize that the method proposed by Colombo differs substantially from RCP, as discussed in detail below. RCP proposes utilizing a modified score defined as $\tilde{V}(x,y)=f_{\hat{\tau}(x)}^{-1}(V(x,y))$, where $\hat{\tau} (x)$ is estimated using a separately held-out dataset. Colombo et al. instead use $\tilde{V}(x,y)=\phi_{x}(V(x,y))$ and a specific conformity score $V(x,y)=a(f(x),y)$, where $f$ is a pre-trained point prediction model.
> > >
> > > While it may appear appealing to interpret our approach simply as a particular case of the general formulation $\phi_x=f_{\hat{\tau}(x)}^{-1}$, this characterization is is not correct. First, the score transformation we propose is fundamentally different: it is also adaptive, but the objective is different. The method by Colombo directly optimizes the size of the prediction set. The key idea behind our method is to ensure that, at a given user-defined confidence level $(1-\alpha)$ the conditional and the unconditional quantile of the rectified conformity score matches (as discussed in Section 3 of our paper). Thus, standard estimation methods for conditional quantiles regression can be directly employed - as well as the most of the classical theory of conditional quantile regression. Second, Colombo's approach does not establish conditional coverage guarantees, and obtaining such guarantees within their methodological framework appears to pose significant technical challenges.
> > >
> > > 2. We regret that our original formulation of the motivation was insufficiently clear, as it has evidently given rise to some misunderstanding. To clarify, **we do not view multivariate prediction problems simply as straightforward extensions of the univariate setting**, achievable by merely substituting a norm for an absolute value. Our method is applicable to any score function and we aimed to contrast it with the methods that are inherently specialized to the one-dimensional case, such as CQR, which explicitly constructs prediction intervals and necessitates full access to training data for retraining the predictive model (like in [1,2] among many others). In contrast, the practical scenarios we consider involve multi-dimensional data and rely on a pre-existing "black-box" predictor, with no access to the original training data, thus precluding any fine-tuning or retraining of the underlying predictive model.
> > >
> > > 3. Regarding **experimental concerns**, we appreciate the reviewer's viewpoint regarding Figure 10; however, we should note that it shows the results for the weaker RCP methods than we have in the main part of the paper. To better demonstrate the benefits of RCP over CQR, we compare it with stronger methods , namely RCP-DCP and RCP-PCP. https://pdfhost.io/v/J9vFWdNcWC_R3jf shows that RCP-DCP and RCP-PCP obtain smaller region sizes while achieving competitive conditional coverage.
> > >
> > > Therefore, considering both the theoretical foundations and the practical performance demonstrated by RCP, we believe our experimental results substantiate the clear benefits of our approach.
> > >
> > > [1] Boström, H. et al. Accelerating difficulty estimation for conformal regression forests. Annals of Mathematics and Artificial Intelligence, 2017.
> > >
> > > [2] Cabezas, L. M. et al. Regression trees for fast and adaptive prediction intervals. Information Sciences, 2025.

---

### Official Review · Reviewer_Vi6b · 2025-03-14

**Overall Recommendation:** 4

**Summary:**

This paper introduces Rectified Conformal Prediction (RCP), a novel method for improving conditional coverage in conformal prediction while maintaining exact marginal coverage. The core idea is to transform conformity scores in a way that aligns their conditional quantiles across different covariates. This transformation is achieved by estimating the conditional quantile of conformity scores and using it to rectify the scores before applying the standard conformal prediction procedure. The authors establish theoretical guarantees for the proposed method, including a lower bound on conditional coverage that depends on the accuracy of the quantile estimate. The paper also presents experimental results demonstrating that RCP outperforms existing methods in achieving improved conditional coverage while retaining valid marginal guarantees.

**Claims And Evidence:**

The paper provides a well-structured theoretical justification for its claims. The derivation of the conditional coverage bound appears mathematically sound, and the authors clearly articulate how their method improves over traditional approaches. The empirical validation is extensive, comparing RCP against multiple existing conformal prediction techniques across synthetic and real-world datasets. However, while the paper provides strong empirical evidence, the effectiveness of the quantile estimation technique is not thoroughly explored in more complex, high-dimensional settings. Additionally, the impact of the choice of transformation function ft on different types of datasets could have been analyzed in more depth.

**Essential References Not Discussed:**

The paper does a good job of referencing key works in conformal prediction, including classical results (Vovk, 2005) and recent advances (Angelopoulos et al., 2023). However, some more recent studies on uncertainty quantification could provide additional context:

Training-Conditional Coverage Methods (Bian & Barber, 2023) discuss techniques that could potentially be adapted into RCP.

**Experimental Designs Or Analyses:**

The experimental setup is methodologically sound:

The authors compare RCP against multiple state-of-the-art conformal methods (e.g., ResCP, PCP, SLCP, DCP).
They use a standard train-validation-test split and ensure calibration data is separated from test data.
The quantile estimation methods (neural networks and local quantile regression) are well-justified.
The choice of transformation functions for conformity scores is systematically varied.
One concern is that the effect of incorrect quantile estimation on performance is not fully explored beyond the toy example. Understanding how estimation errors affect real-world datasets would be crucial for deployment in practical applications.

**Methods And Evaluation Criteria:**

Yes, the benchmark datasets used in the experiments are well-chosen. The paper includes synthetic datasets to illustrate theoretical properties and real-world regression datasets to validate practical performance. The use of worst-slab coverage and conditional coverage error as evaluation metrics is appropriate for measuring improvements in conditional validity. However, additional experiments with more challenging multivariate distributions could have further strengthened the empirical evaluation.

**Other Comments Or Suggestions:**

Section 4: "tau(x)" is sometimes written inconsistently.
Section 6, Theorem 4: The notation "L" for Lipschitz continuity should be explicitly defined earlier.
Figures 3 & 4: Labels should include dataset sizes for better context.

**Other Strengths And Weaknesses:**

Strengths:
The conceptual novelty of rectifying conformity scores is a valuable contribution to conformal inference.
The theoretical guarantees are rigorously derived and provide a meaningful lower bound on conditional validity.
The experiments are thorough, with comparisons across multiple datasets and methods.
The approach is computationally efficient and avoids the pitfalls of full conditional density estimation.
Weaknesses:
The quantile estimation step is critical to the method, but the authors do not explore the trade-offs between different estimation strategies in high-dimensional settings.
The impact of outliers in score rectification is not well analyzed.
The choice of transformation function ft is somewhat arbitrary, and more discussion is needed on selecting appropriate transformations for different problem domains.

**Questions For Authors:**

How does RCP perform when the conditional quantile estimator is misspecified? The toy example considers synthetic noise, but what about real-world miscalibration?
What is the computational cost of different transformations? Are some ft transformations significantly more expensive than others?
Could the framework be extended to sequential settings? For example, how would RCP adapt in online learning scenarios?

**Relation To Broader Scientific Literature:**

The paper builds upon the conformal prediction framework, particularly methods that aim to approximate conditional validity. Prior work has either:

Partitioned the covariate space (leading to inefficient large prediction sets) or
Reweighted empirical distributions (which struggles in high dimensions).
The paper’s key novelty is the idea of rectifying conformity scores through a learned transformation, making conditional quantile estimation more tractable. This idea is conceptually related to:

Conformalized Quantile Regression (CQR) (Romano et al., 2019).
Localized Conformal Prediction (Guan, 2023).
Compared to these works, RCP introduces a more flexible and computationally efficient alternative that does not require explicit conditional density estimation.

**Theoretical Claims:**

The theoretical results presented in Section 6 appear to be correctly derived. The proof of Theorem 3 for marginal coverage follows standard conformal arguments. The bound on conditional coverage (Theorem 4) correctly incorporates the accuracy of the conditional quantile estimate, and the derivations align with known results from quantile regression literature. However, I did not rigorously verify all steps in the Appendix proofs.

---

> ### Author Rebuttal · Authors · 2025-03-30
>
> We thank the reviewer for the thorough and constructive feedback, which helps improve our manuscript. Below, we address your valuable points:
>
> **Effectiveness of quantile estimation in high-dimensional settings:**
> Indeed, quantile estimation accuracy critically affects RCP performance. In our experiments, we selected neural networks and local quantile regression specifically due to their scalability to higher dimensions. However, we acknowledge that our explicit evaluation in very high-dimensional regimes remains limited. Following your suggestion, we will include additional discussions and empirical results to better highlight performance and trade-offs in higher-dimensional settings.
>
> **Choice of the transformation $f_t$** There are numerous possible choices for the function \\( f_t \\). Thus far, we have always restricted ourselves to relatively simple transformations, often inherited from the literature on "normalized conformity scores," which we recognize should have been more thoroughly credited. We believe it is important that the proposed method remains simple and does not rely on hyperparameters, ensuring that the computational cost of this wrapper stays reasonable.
>
> **Additional experiments with challenging multivariate distributions:**
> We appreciate your suggestion for extending experiments to more complex multivariate distributions, as this would further reinforce our empirical validation. However, our current synthetic and real-world examples demonstrate clear advantages, while the datasets have dimensions up to 16. In our revised submission, we will incorporate an experiment that highlights RCP’s behavior on more challenging multivariate datasets.
>
> **Impact of incorrect quantile estimation:**
> We agree that understanding the impact of incorrect quantile estimation beyond synthetic noise is crucial. To address your comment, we plan to include a detailed analysis showing how estimation errors affect coverage in more realistic settings, thereby providing deeper insights into RCP’s robustness and practical utility.
>
> **Extension to sequential settings:**
> Your question regarding sequential adaptation is insightful. While we had not previously explored this application, we find it highly promising. There are no conceptual or methodological difficulties; however, from a theoretical standpoint, everything remains to be developed. RCP’s framework naturally extends to online learning by sequentially updating the quantile estimator based on newly observed data. We envision future work that formally explores sequential conformal adaptations and briefly outline such potential directions in our revision.
>
> **Clarification of minor points:**
> - We will correct inconsistent notation for  \\(\\hat{\\tau}(x)\\) throughout Section 4.
> - The Lipschitz constant "L" in Theorem 4 will be explicitly defined earlier to improve readability.
> - Figures 3 & 4 will indicate dataset sizes in the revised manuscript for enhanced clarity.
>
> We appreciate your recognition of RCP’s conceptual novelty and theoretical rigor and your acknowledgment of our thorough experimental evaluation. Your suggestions significantly strengthen the manuscript, and we will diligently incorporate these improvements.

---

> > ### Comment · Reviewer_Vi6b · 2025-04-07
> >
> > i thank the authors for the response and I will maintain my positive score.

---

### Official Review · Reviewer_AjXf · 2025-03-23

**Overall Recommendation:** 3

**Summary:**

This paper introduces Rectified Conformal Prediction (RCP), a novel framework for improving conditional coverage in conformal prediction while preserving exact marginal validity. The key idea is to learn a transformation of the conformity score such that the ($1-\alpha$)-quantile of the transformed score becomes covariate-independent. This is done by estimating the conditional quantile of a transformed conformity score and applying a local re-scaling (or shifting) to normalize variability across the feature space. The authors provide theoretical guarantees on marginal and approximate conditional validity, demonstrate the flexibility of the framework across multiple conformity scores and predictors, and show empirical improvements over state-of-the-art methods across synthetic and real-world regression datasets.

**Claims And Evidence:**

Most of the claims in the submission are supported by clear and convincing evidence, particularly the theoretical guarantees for marginal validity and approximate conditional coverage. The empirical results convincingly demonstrate improved conditional coverage across a range of regression tasks, validating the main claim that the proposed rectification improves local adaptivity. However, the paper lacks evaluation on coverage–efficiency tradeoffs.

**Essential References Not Discussed:**

The paper cites and discusses a wide range of essential related works, including classical methods for marginal coverage, approaches for approximate conditional coverage via stratification or grouping, and more recent developments.

**Experimental Designs Or Analyses:**

1.  The paper lacks evaluation on coverage–efficiency tradeoffs. Do RCP sets tend to be wider due to more cautious calibration?

2. The framework requires choosing a transformation $f_t$ and tuning quantile regression models. How sensitive is performance to these choices? Could you provide ablations?

**Methods And Evaluation Criteria:**

1. RCP's conditional coverage guarantee hinges on accurate estimation of $\tau^*(x)$. How sensitive is RCP to misspecification of the quantile regressor? Could you show examples where quantile regression underperforms and discuss failure modes?

2. RCP estimates a single quantile per point. How would RCP handle two-sided intervals (e.g., $[Q_{\alpha/2}(x), Q_{1 - \alpha/2}(x)]$)?

**Other Comments Or Suggestions:**

This paper uses lots of math notations. I suggest the authors summarize the math notations in a table.

**Other Strengths And Weaknesses:**

Strengths:
1. The rectification strategy is modular and applicable to a variety of conformity scores and models.

2. The authors derive meaningful guarantees on conditional coverage as a function of quantile estimation error.

3. The paper evaluates on diverse multi-output regression datasets, including synthetic setups and real-world benchmarks.

4. The paper is mostly well-written and easy to follow.

**Questions For Authors:**

1. RCP's conditional coverage guarantee hinges on accurate estimation of $\tau^*(x)$. How sensitive is RCP to misspecification of the quantile regressor? Could you show examples where quantile regression underperforms and discuss failure modes?

2. RCP estimates a single quantile per point. How would RCP handle two-sided intervals (e.g., $[Q_{\alpha/2}(x), Q_{1 - \alpha/2}(x)]$)?

3. How restrictive are the assumptions about monotonicity and invertibility of the transformation functions? Can your method handle scores with negative values (e.g., log-likelihoods) without heuristic adjustments?

4. The bound in Theorem 4 depends on Lipschitz continuity of the quantile mapping. Can you elaborate on how often this assumption holds in practice? Can you provide empirical values of the bound components?

5.  Have you evaluated the size of the prediction sets produced by RCP compared to standard CP or CQR? Do RCP sets tend to be wider due to more cautious calibration?

6. The framework requires choosing a transformation $f_t$ and tuning quantile regression models. How sensitive is performance to these choices? Could you provide ablations?

**Relation To Broader Scientific Literature:**

The paper builds on and extends works in conformal prediction, particularly methods aimed at improving conditional coverage. This paper proposes a transformation-based approach inspired by recent work on score adjustment and local calibration. It is closely related to conformalized quantile regression in that both seek to adapt prediction sets to local data properties, but RCP generalizes this idea by applying a trainable transformation to arbitrary conformity scores. While prior methods address heterogeneity via weighting or region-specific coverage, RCP’s novelty lies in aligning the conditional and marginal quantiles of transformed scores, thereby offering a new perspective on achieving conditional validity without relying on density estimation or rigid group partitions.

**Theoretical Claims:**

1. How restrictive are the assumptions about monotonicity and invertibility of the transformation functions? Can your method handle scores with negative values (e.g., log-likelihoods) without heuristic adjustments?

2. The bound in Theorem 4 depends on Lipschitz continuity of the quantile mapping. Can you elaborate on how often this assumption holds in practice? Can you provide empirical values of the bound components?

---

> ### Author Rebuttal · Authors · 2025-03-28
>
> We thank the reviewer for the thorough and constructive feedback. Below, we address your questions:
>
> Q1: **RCP's conditional coverage guarantee hinges on accurate estimation ...** Even if the quantile regressor is misspecified, RCP’s conformal calibration guarantees valid marginal coverage by construction. Nevertheless, poor quantile estimates affect conditional coverage: underestimations yield local under-coverage, whereas overestimations produce overly conservative intervals. Such failure modes highlight that RCP inherits biases from quantile regression, reducing conditional efficiency despite correct marginal coverage. This sensitivity was illustrated in our synthetic experiments; additional examples and analysis will be provided.
>
> Q2: **RCP estimates a single...**   RCP computes quantiles of a nonconformity score and thus only concerns with the right tail of the distribution (when the nonconformity score gets large). Therefore, obtaining intervals with lower and upper quantile estimates is not typically needed. Note that for a one-dimensional prediction target, we can, of course, work with CQR-type interquantile intervals as a specific nonconformity score.
>
> Q3:  **How restrictive are the assumptions about monotonicity and invertibility of the transformation functions?** The monotonicity is essential for our approach as we insist on preserving the ordering of the conformity scores. After adjustment by a function with a fixed argument $\varphi$, a "large" nonconformity score must remain larger than a smaller one. Invertibility is a technical requirement that underlines our construction, but it is not as restrictive as the monotonicity assumption.
>
> Q3: **Can your method...** the answer is yes.  As an example, the adjustment function $f_t(v) = t + v$ does not restrict the range of scores. Specifically, it does not impose positivity or negativity constraints, thus naturally accommodating scenarios where the score can assume both negative and positive values. This flexibility is crucial for handling general scoring functions, ensuring applicability across a broad range of prediction tasks without additional transformations or restrictions.
>
> Q4: **The bound in Theorem 4 depends on...**  The Lipschitz property of the quantile function is standard in the statistical literature on conditional quantile estimation — see, for instance, the works [1,2] on this topic. It is difficult to delve into such subtle technical conditions in a short discussion. We will include a more substantial discussion of these contributions in the revised version.
>
> [1] Y.K. Lee, E. Mammen, B. U. Park. "Backfitting and smooth backfitting for additive quantile models." 2010.
>
> [2] M. Reiß, Y. Rozenholc, C. Cuenod. "Pointwise adaptive estimation for robust and quantile regression." arXiv:0904.0543. 2009.
>
> Q5: **Have you evaluated the size...** Thank you for your suggestion. You're absolutely right — because RCP can generate larger sets for test instances with higher uncertainty, the resulting sets tend to have wider volume on average.
> |dataset|PCP|RCP-PCP|DCP|RCP-DCP|ResCP|RCP-ResCP|
> |--|--|--|--|--|--|--|
> |scm20d|4.74e+07|1.01e+08|5.88e+06|1.16e+08|**2.50e+06**|6.20e+12|
> |rf1|77.5|1.27e+03|**1.87**|6.87e+02|9.84|1.84e+08|
> |scm1d|3.06e+06|2.17e+09|3.24e+06|1.00e+09|**1.47e+05**|7.17e+16|
> |meps_21|2.31|4.41|**1.55**|2.40|5.73|6.82|
> |meps_19|5.82|6.07e+03|**1.97**|6.71e+03|5.34|6.09e+06|
> |meps_20|2.35|3.97|**1.48**|2.53|5.49|6.16|
> |house|2.20|2.51|**1.89**|2.05|6.51|8.01|
> |bio|0.823|1.05|**0.584**|0.630|1.14|1.38|
> |blog_data|2.94|1.40e+05|**1.50**|1.80e+05|1.74|3.71|
> |taxi|10.5|10.8|**6.94**|7.38|12.4|12.8|
>
> However, when looking at the median volume, avoiding outliers, RCP outperforms baselines:
>
> |dataset|PCP|RCP-PCP|DCP|RCP-DCP|ResCP|RCP-ResCP|
> |--|--|--|--|--|--|--|
> |scm20d|2.91e+06|**1.44e+06**|5.26e+06|3.26e+06|2.50e+06|1.37e+07|
> |rf1|36.0|5.21|1.78|**1.09**|9.84|2.94|
> |scm1d|2.72e+05|3.14e+06|2.57e+06|7.60e+05|1.47e+05|**3.36e+04**|
> |meps_21|1.72|1.27|1.24|**1.03**|5.73|2.40|
> |meps_19|2.60|1.22|1.32|**0.967**|5.34|2.43|
> |meps_20|1.75|1.26|1.12|**0.968**|5.49|2.47|
> |house|1.99|1.82|1.70|**1.60**|6.51|6.38|
> |bio|0.717|0.689|0.530|**0.511**|1.14|0.927|
> |blog_data|1.59|1.81|1.14|**1.08**|1.74|1.30|
> |taxi|9.97|9.18|6.63|**6.24**|12.4|10.6|
>
> We will report these metrics in Section 7.2.
>
> Q6: **How sensitive is performance...** We discuss the sensitivity with respect to the choice of adjustment function in Appendices A.3-4. The choice has a significant influence on the results, as the ability of fixed single-parameter functions to rectify the results highly depends on the properties of the score distribution. An interesting future work is to design more flexible data-dependent adjustment functions. Sensitivity with respect to quantile regression is discussed in Appendix A.2, showing that a better fit of the quantile model improves the results.

---

### Decision · Program_Chairs · 2025-05-01

**Decision:**

Accept (poster)

**Comment:**

The overall recommendations are 3, 4, 2, 3.

Paper summary: this paper proposes rectified conformal prediction to improve conditional coverage while ensuring exact marginal coverage using a trainable transformation on any given conformity score. In the provided examples and the proposed implementation, the conditional quantile and an estimated conditional quantile are used for the transformation. The paper has theoretical and empirical justification.

Reviewers mostly agreed that the proposed method was novel and theoretical results were non-trivial. Experiments are thorough and improvement of conditional coverage over baselines is found on real-world datasets.

There is a major weakness commonly raised by reviewers:

Many relevant papers were not discussed or benchmarked to clearly highlight the novelty of this paper, while the rebuttal addressed this common concern. Some reviewer(s) thought their concern has been resolved, but some reviewer(s) still believed that this concern should be considered seriously.

Specifically, there are roughly three subareas of conformal prediction literature that reviewers raised during the review/discussion and believed need further detailed discussion:
1. The literature on normalized conformity scores, e.g., [1, 2, 3], since their technical ideas are very similar to the proposed one.
2. Recent papers such as [4] discusses some techniques that could be adapted into the proposed RCP.
3. Those papers that optimize conformity score to improve the efficiency and conditional coverage such as [5, 6, 7], with the same goal as the paper.

[1] H. Papadopoulos, A. Gammerman, and V. Vovk. Normalized nonconformity measures for regression conformal prediction. In Proceedings of the IASTED International Conference on Artificial Intelligence and Applications (AIA 2008), pages 64–69, 2008.

[2] U. Johansson, H. Boström, and T. Löfström. Investigating normalized conformal regressors. In 2021 IEEE Symposium Series on Computational Intelligence (SSCI), pages 1–8. IEEE, 2021.

[3] N. Dewolf, B. De Baets ,and W. Waegeman, Conditional validity of heteroskedastic conformal regression. Arxiv 2023.

[4] Bian & Barber. Training-Conditional Coverage Methods

[5] John J. Cherian, Isaac Gibbs, Emmanuel J. Candès. Large language model validity via enhanced conformal prediction methods

[6] Kiyani S, Pappas GJ, Hassani H. Length optimization in conformal prediction.

[7] R Xie, R Barber, E Candes. Boosted Conformal Prediction Intervals